# Heterogeneity analysis provides evidence for a genetically homogeneous subtype of bipolar-disorder

Caroline C. McGrouther[1], Aaditya V. Rangan[1]*, Arianna Di Florio[2], Jeremy A. Elman[3], Nicholas J. Schork[4], John Kelsoe[5], Bipolar Disorder Working Group of the Psychiatric Genomics Consortium[¶]

1 Courant Institute of Mathematical Sciences, New York University, New York, NY, United States of America, 2 School of Medicine, Division of Psychological Medicine and Clinical Neurosciences, Cardiff University, Cardiff, United Kingdom, 3 Department of Psychiatry, University of California San Diego, San Diego, CA, United States of America, 4 The Translational Genomics Research Institute, Quantitative Medicine and Systems Biology, Phoenix, AZ, United States of America, 5 Department of Psychiatry, University of California San Diego, La Jolla, CA, United States of America

¶ Membership of Bipolar Disorder Working Group of the Psychiatric Genomics Consortium is provided in the Acknowledgments.
* avr209@nyu.edu

**Data Availability Statement:** There are no primary data in the paper; all of the code is available at https://github.com/adirangan/ The data which is

## Abstract

### Background

Bipolar Disorder (BD) is a complex disease. It is heterogeneous, both at the phenotypic and genetic level, although the extent and impact of this heterogeneity is not fully understood. One way to assess this heterogeneity is to look for patterns in the subphenotype data. Because of the variability in how phenotypic data was collected by the various BD studies over the years, homogenizing this subphenotypic data is a challenging task, and so is replication. An alternative methodology, taken here, is to set aside the intricacies of subphenotype and allow the genetic data itself to determine which subjects define a homogeneous genetic subgroup (termed 'bicluster' below).

### Results

In this paper, we leverage recent advances in heterogeneity analysis to look for genetically-driven subgroups (i.e., biclusters) within the broad phenotype of Bipolar Disorder. We first apply this covariate-corrected biclustering algorithm to a cohort of 2524 BD cases and 4106 controls from the Bipolar Disease Research Network (BDRN) within the Psychiatric Genomics Consortium (PGC). We find evidence of genetic heterogeneity delineating a statistically significant bicluster comprising a subset of BD cases which exhibits a disease-specific pattern of differential-expression across a subset of SNPs. This disease-specific genetic pattern (i.e., 'genetic subgroup') replicates across the remaining data-sets collected by the PGC containing 5781/8289, 3581/7591, and 6825/9752 cases/controls, respectively. This genetic subgroup (discovered without using any BD subtype information) was more prevalent in Bipolar type-I than in Bipolar type-II.

analyzed can be requested from the Psychiatric Genomics Consortium at: https://pgc.unc.edu/.

**Funding:** AVR and NJS are supported in part in part by the National Institutes of Health grant: U19AG023122. The funders did not play any role in the study design, data collection and analysis, decision to publish or preparation of this manuscript.

**Competing interests:** The authors have declared that no competing interests exist.

## Conclusions

Our methodology has successfully identified a replicable homogeneous genetic subgroup of bipolar disorder. This subgroup may represent a collection of correlated genetic risk-factors for BDI. By investigating the subgroup's bicluster-informed polygenic-risk-scoring (PRS), we find that the disease-specific pattern highlighted by the bicluster can be leveraged to eliminate noise from our GWAS analyses and improve risk prediction. This improvement is particularly notable when using only a relatively small subset of the available SNPs, implying improved SNP replication. Though our primary focus is only the analysis of disease-related signal, we also identify replicable control-related heterogeneity.

## Background

### Overview

Bipolar disorder (BD) is a brain disorder characterized by shifts in mood, energy and attention/focus [1]. BD affects roughly 50 million people across the world, with a mean age of onset of 20 years and an estimated lifetime prevalence of $\sim 1\%$ [2–5]. BD is also highly heritable [6], with heritability estimates of 40% or higher [7–11] and evidence of increased risk when family-members exhibit other psychiatric disorders [7, 10, 11].

There is growing consensus that BD is heterogeneous, both at the phenotypic and genetic level [12–23]. For example, diagnostic systems usually consider at least two subtypes of bipolar disorder: bipolar I and bipolar II. The diagnostic criteria for bipolar I require the presence of at least one manic episode, while those for bipolar II require at least one hypomanic and one major depressive episode [1]. Response to medication (such as lithium) is highly heterogeneous across patients, and genetic predictors of drug-response have been difficult to clearly determine and replicate [24–28].

The high degree of heterogeneity for BD at the clinical and phenotypic level may make it more difficult to identify genetic risk-factors for BD. To briefly summarize: while the overall heritability of BD is estimated at $\sim 40\%$, the overall single-nucleotide-polymorphism (SNP) heritability is only $\sim 18.6\%$ [29], which is moderate when compared to many other psychiatric and neurological disorders [6, 30–38]. Recent genome-wide association studies (GWASs) have been used to identify several (i.e. $\sim 100$) independent loci associated with BDI and BDII, with the overall variance explained by SNPs reaching $\sim 15 - 18\%$ [29]. However, many of the loci that seem promising in one cohort fail to replicate in other cohorts [23, 39, 40]. Studies attempting to uncover gene-environment interactions in BD have also encountered challenges finding replicable signals [41–45].

Rather than focusing on small sets of loci, one can also consider collections of SNPs which individually may not be of genome-wide significance. Along this vein, Polygenic-risk-scores (PRSs), which are usually weighted sums of genetic variants, have been used to summarize the genome-wide risk for BD [46]. These PRSs may provide an estimate of overall risk and/or severity: those individuals with PRSs in the top 90% were 3.62 times more likely to be a case than those with average PRSs. These PRSs also contain information regarding multiple phenotypic traits, including the risk of other psychiatric disorders, psychopathology, educational attainment and more [47–55]. Despite these successes, to the best of our knowledge, no individual PRS has yet been able to explain a large fraction of the variation between the main bipolar subtypes.

The high degree of heterogeneity within BD poses a challenge to understanding its etiology and developing new interventions. Ultimately, a comprehensive depiction of the landscape of BD will involve clear descriptions of the heterogeneity at the phenotypic level, as well as at the genetic level.

To date, the main research efforts aimed at understanding the genetic heterogeneity underlying BD have focused on (i) increasing the power of BD meta-GWAS, (ii) running subphenotypic-specific meta-GWAS, and (iii) performing pathway-specific analyses [56–58]. These research efforts are non-trivial and in some cases require insights we do not yet have. Generally speaking, recruiting, assessing, and genotyping new subjects is expensive; there is often a trade-off between the quantity of subjects that can be recruited and the 'quality' or accuracy with which their data is processed. For example, one promising resource for genotyped data is 23andMe, but many of the data-sets available through this resource rely on self-reported diagnoses [59]. Consequently, any synchronization effort involves the integration and harmonization of data collected using different phenotypic instruments or genotyping methods and may inadvertently introduce non-disease-related signal. Furthermore, in many cases, the relevant subphenotypic information was not collected at all, forcing interested researchers to contact prior participants or lose those data points entirely. Finally, even when promising results are obtained, it is not always easy to find an appropriate replication sample [60]. Since we do not yet know which trait or combination of subphenotypic traits (if any) is responsible for BD genetic heterogeneity, it is not always clear how best to proceed.

## Contribution

Ultimately, we seek to investigate the genetic heterogeneity of BD by using an approach which does not require the user to provide pathways or subphenotypes. As described below, we introduce a methodology which first uses the genotyped data to identify a genetic subgroup within BD, and then uses that genetic subgroup for downstream analyses (in this case risk prediction). To briefly summarize: we use a covariate-corrected biclustering algorithm to search for statistically significant biclusters comprising subsets of BD cases which exhibit disease-specific patterns of differential-expression across subsets of SNPs. In this study we find one statistically-significant disease-specific structure, which is limited to only a fraction of the case-subjects. These case-subjects collectively exhibit a shared pattern of differential-expression—i.e., a form of genetic homogeneity—which is not shared by the other BD-cases nor by the control-subjects; we refer to this bicluster as a 'genetic subgroup'. We then demonstrate that this genetic subgroup is useful for risk-prediction.

In more detail, our analysis begins by collecting data within which to search for genetic subgroups of BD. As members of the Psychiatric Genomics Consortium (PGC), we had access to the raw genotypes of $\sim 18K$ BD cases and $\sim 30K$ controls. This data was generated by 27 studies and genotyped on a variety of platforms (OMEX, Affymetrix, Illumina). When the PGC analyzed this data [20], they synchronized the data using imputation. We were not certain how imputation might impact the potentially subtle relationships between BD cases, and therefore decided to limit our analysis to the available raw genotyped data [60]. This choice to limit ourselves to raw genotyped data placed constraints on our choices for the training and testing data sets, as the various genotyping platforms types emphasize different SNP sets (see Fig 1).

In order to minimize batch-effects and reduce the chances of spurious false-positives, we chose to initially focus our primary analysis on a relatively large curated study from the Bipolar Disorder Research Network (BDRN) comprising raw genotyped data collected across 2524 BD cases and 4106 controls (OMEX platform) [18]. We use this BDRN study as our training-arm,

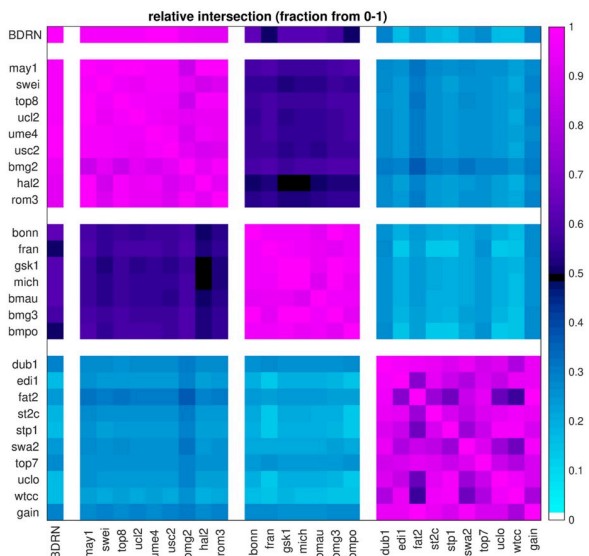 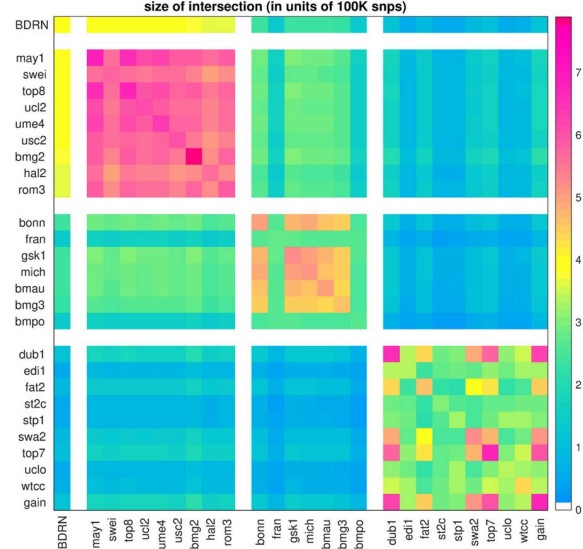

**Fig 1.** In this figure we illustrate the absolute (right) and relative (left) snp overlap between the studies available to us. The relative-overlap is calculated using the Szymkiewicz–Simpson coefficient (i.e., the overlap-coefficient between sets $X$ and $Y$ is $|X \cap Y|/\min(|X|, |Y|)$). Guided by the relative-overlap and genotyping platform used, we divided the studies into four arms (shown along the coordinate axes). The first arm contains only the single 'BDRN' data-set, which we use as a training/discovery set to search for heterogeneity (see Methods). We reserve the remaining studies (organized into three arms) for replication. Note that the training-set overlaps strongly with arm-2, and less strongly with arm-3 and arm-4. The magnitude of this overlap will constrain how faithfully any patterns of differential-expression found in arm-1 can possibly manifest within the other arms (see Figs 3–5).

and set aside the remaining independent data for subsequent replication analyses (i.e., our replication-arms). We grouped all the BD cases in our training-arm together and searched within the training-arm for any subsets of subjects which exhibited a distinct genetic signature (i.e., differential expression) across a subset of SNPs. Any such subset of subjects along with the associated subset of differentially-expressed SNPs is referred to as a 'bicluster', or a 'genetic subgroup'.

As described in [61, 62], many commonly used biclustering approaches suffer from two methodological issues. First, a bicluster that is found within the case-population may not be disease-related, as a similar signal may be found within the control-population (e.g., a bicluster representing non-disease-specific heterogeneity). Second, many biclustering algorithms proceed under the assumption that biclusters exist, often identifying 'false-positive' structures that are not statistically-significant.

To address these issues we searched for biclusters using the 'half-loop' algorithm of [63, 64]. As described in [64], this algorithm ensures that the pattern of differential-expression within the bicluster is *not* similarly present within the control-population, reducing the likelihood that we highlight structures unrelated to disease status. Second, the half-loop algorithm uses a permutation-test to estimate the p-value of each bicluster found, allowing us to test against the null hypothesis that no bicluster exists. Finally, the half-loop algorithm also allows us to correct for other covariates, such as proxies for genetic-ancestry (see Methods). While our approach is much simpler than some of the more recent machine-learning approaches, our biclusters are directly associated with subject- and SNP-subsets, which can be directly interpreted and assessed for homogeneity and/or used in downstream analyses.

Using the relatively conservative half-loop method mentioned above, we found strong evidence for genetic heterogeneity. We discovered one bicluster which is statistically significant and which replicates in all three other data-sets. This primary bicluster was enriched for (but

not completely driven by) BDI over BDII. After removing this bicluster we saw further evidence of residual heterogeneity, but our training data-set was not sufficiently powered to clearly identify a secondary bicluster.

We then assessed the role of our bicluster in risk-prediction. We found that the subset of case-subjects highlighted by the bicluster can be used to improve the performance of a PRS. This advantage was more pronounced when (i) the SNPs included in the PRS were limited to those of high estimated significance, and (ii) the case-population was limited to those diagnosed with BDI. These observations suggest that focusing on genetically identifiable subgroups of BD-subjects might improve overall risk-prediction and enhance replication across the top SNPs.

Finally, we also ran a simple gene-set over-representation analysis, revealing that the genetic subgroup identified above (i.e., the bicluster) is significantly enriched for many pathways associated with neuronal development and maintenance.

In summary, we find strong evidence for the genetic-heterogeneity of BD in the form of a bicluster. Notably, BD subphenotype information was not required to identify this signature, nor were rare-variants (i.e., we relied only on common SNPs with maf greater than 25% within the training-arm). The signature of this bicluster has the potential to refine downstream analyses (e.g., improving genome-wide risk-prediction), and the associated gene-enrichment suggests an association with certain mechanisms of neuronal development.

## Methods

In this section we describe several aspects of our methodology. An even more detailed description, including an outline of the steps involved and the considerations we made along the way, is available in S1 Text.

### Data

We make use of data from 27 of the cohorts described in [20]. These cohorts have been curated as described in [20] and its supplementary information, and include de-identified subjects from several countries in Europe, North America and Australia, totaling over 18000 cases and 29000 controls of European descent. Case-subjects were required to meet international consensus criteria (DSM-IV, ICD-9, or ICD-10) for a lifetime diagnosis of BD established using structured diagnostic instruments from assessments by trained interviewers, clinician-administered checklists, or medical record review. Control-subjects in most samples were screened for the absence of lifetime psychiatric disorders, as indicated. For each of the 27 cohorts, we had access to both the raw genotypes and the imputed data generated by Stahl et al. using the 1000 Genomes (1KG) European reference-panel (see [20]).

Due to the details of our heterogeneity analysis (described further below), we make three additional choices. First, for our primary analysis we use only the raw genotyped data within each cohort, but not the imputed data. This is because we want to avoid any concerns of spurious correlations that might arise from imputation [60]. Second, when running our biclustering algorithm we do not explicitly correct for linkage-disequilibrium (LD) between genotyped SNPs at the level of the data-set itself (e.g., by eliminating SNPs in strong LD with other SNPs). Instead, we implicitly correct for LD within our biclustering algorithm by contrasting cases against controls. Third, it is typically quite difficult to reliably detect signal associated with rare variants (i.e., SNPs with a low minor-allele-frequency, a.k.a. 'maf'), especially when the power of the data-set is low. This difficulty is compounded when searching for heterogeneity, as the effective sample-size (e.g., the number of subjects in a bicluster) is further reduced—often only a fraction of the total subject-population [64]. Thus, in order to avoid spurious results

associated with rare-variants, we limit our analysis to common variants (i.e., SNPs with maf greater than 25%). This high maf-threshold has the added benefit that the signals that we do find are described in terms of common variants, which will hopefully be easier to access in future studies.

As shown in Fig 1, the common genotyped SNP-overlap between the cohorts varies significantly. Cohorts that were genotyped using similar platforms tend to have large SNP-overlaps, while those genotyped on different platforms tend to have smaller SNP-overlaps. After clustering the cohorts by platform (and removing any duplicate subjects across cohorts) we defined four 'arms', as shown along the axes in Fig 1. Arm-1 consists of the single cohort labeled 'BDRN' (2524 cases, 4106 controls, OMEX). Arm-2 includes cohorts 'may1' through 'rom3' (5781 cases, 8289 controls, OMEX). Arm-3 includes cohorts 'bonn' through 'bmpo' (3581 cases, 7591 controls, Illumina). Arm-4 includes cohorts 'dub1' through 'gain' (6825 cases, 9752 controls, Affymetrix).

The first arm (comprising the single cohort 'BDRN') is relatively large and collected within the UK, comprising case-subjects of European descent over the age of 17 (see [18, 65, 66] for details). As a result, we expect this cohort to be less susceptible to spurious heterogeneity associated with batch-effects, and we use this cohort as a 'training' or 'discovery' arm, reserving the other three independent data-sets for validation (i.e., 'replication' arms). This training-arm has a large SNP-overlap of $\sim 85\%$ with arm-2, and a smaller SNP-overlap with arms 3 and 4 (i.e., $\sim 50\%$ and $\sim 30\%$, respectively). Correspondingly, we expect that any signal involving a multi-SNP-pattern found in arm-1 will only have an opportunity to replicate strongly in arm-2, and will not have the opportunity to replicate as strongly in arms 3 and 4 (as we will have fewer SNPs to use for validation).

**Ethics statement.** We first obtained access to this data on 2013–11-25, and we have never had access to any information that could identify individual participants during or after data collection. This study was approved by the institutional review boards (IRBs) at University of California, San Diego as well as New York University. Because all the data we were working with had been de-identified, both IRBs certified our study as exempt from review and continuing review. The metadata for the subjects included genome-wide principal-components, which we used as a proxy for ancestry and corrected for in our primary- and secondary analyses, as described below. The metadata also included sex, whose effect we assessed a-posteriori (see Fig 17 in S1 Text). We did not have access to confounding variables such as socioeconomic status, nutrition, environmental exposures, or other similar factors, and could not correct for these in our analysis.

## Correcting for ancestry

We use the genome-wide principal-components calculated by Stahl et al. to assess relatedness and correct for ancestry. Of the first 20 principal-components, denoted $\{U_1, \ldots, U_{20}\}$, Stahl et al. determined that the first six principal-components ($U_1$-$U_6$) and $U_{19}$ showed significant correlation with the main phenotype across the studies considered in [20].

In the discovery phase of our analysis we are restricted to the training-arm (arm-1). For this sample we use an F-test applied to a nested logistic model, which selects only the first two principal-components (i.e., $U_1$ and $U_2$) as significantly related to case-control status in arm-1. Therefore, to mimic the analyses one might conduct with access only to arm-1, we correct our biclustering algorithm for these two principal-components, under the assumption that they are a proxy for ancestry.

In all but this initial biclustering analysis on arm-1, we remain consistent with [20] and correct for principal-components $U_1$ through $U_6$, as well as $U_{19}$. This includes both the calculation

of **A U C** in the subsequent replication studies (e.g., $A(i)$ and $A'(i)$ in Fig 3) as well as the PRS-analysis described below.

## Biclustering

For our initial biclustering of arm-1 we use the half-loop method of [64]. To briefly summarize the method, we first introduce some notation. Assume that the data-set contains $M_D$ case-subjects, and $M_X$ control-subjects, each measured across $N$ allele-combinations (note, each SNP is associated with three allele combinations: heterozygous and homozygous dominant and recessive). We denote the array of case-subjects by $D$, with $D(j_D, k)$ referring to allele-combination-$k$ in case-subject-$j_D$. Similarly, we denote the array of control-subjects by $X$, with $X(j_X, k)$ referring to allele-combination-$k$ in control-subject-$j_X$. We'll use the generic subject-index $j$ to refer to both the $j_D$ and the $j_X$.

In its most basic form, the half-loop algorithm proceeds as follows:

**Step-0** First we load/initialize the data-arrays $D$ and $X$.

**Step-1** For each case $j_D$ and allele-combination $k$, we measure the fraction of other cases in $D$ which share that allele-combination, denoted by $[D \leftarrow D](j_D, k)$. Similarly, we measure the fraction of controls in $X$ which share that allele-combination, denoted by $[D \leftarrow X](j_D, k)$. The difference between these two values, denoted by $Q(j_D, k) = [D \leftarrow D](j_D, k) - [D \leftarrow X](j_D, k)$ is a measure of differential-expression.

**Step-2** After calculating $Q(j_D, k)$, we form the 'row-scores' $Q^{\text{row}}(j_D) = \Sigma_k Q(j_D, k)$, as well as the 'column-scores' $Q^{\text{col}}(k) = \sum_{j_D} Q(j_D, k)$ and the 'trace' $\bar{Q} = \sum_{j_D, k} Q(j_D, k)$. The row- and column-scores measure how strongly each case-subject and allele-combination contribute to the trace, which is itself a measure of the overall differential-expression exhibited between $D$ and $X$.

**Step-3** We remove a small fraction of case-subjects and allele-combinations from $D$ with the lowest row- and column-scores.

**Step-4** We return to Step-1, iterating until there are no more case-subjects within $D$.

The algorithm proceeds iteratively; at each iteration $i$ we remove a small fraction $\gamma$ of the remaining case-subjects and allele-combinations. In this analysis we choose $\gamma = 0.5^8 \sim 0.004$, which is sufficiently small that we expect statistical convergence of the algorithm's accuracy (see Fig 32 in supplementary section 7.3 in [64]). After each iteration $i$, a subset $\mathcal{J}(i)$ comprising $M(i)$ case-subjects and a subset $\mathcal{K}(i)$ comprising $N(i)$ allele-combinations remain, together forming an $M(i) \times N(i)$ sub-array $D(i)$ of the original $D$. If the case-array $D$ were to contain a bicluster with a sufficiently strong signal, then the rows and columns of that bicluster would be retained until the end, with the other rows and columns eliminated earlier.

This half-loop method has detection-thresholds similar to spectral-clustering and message-passing [67, 68], but has several additional useful features. First, the half-loop method allows us to search for disease-specific heterogeneity by directly correcting for control-subjects. This case-control correction also motivates the null-hypothesis H0 described below; the permutation-test allows us to avoid spurious structures that are unrelated to the disease-label. Second, the half-loop scores in Step-1 allow us to (implicitly) correct for linkage-disequilibrium (LD). More specifically, subsets of SNPs which are in equally strong LD in both the case- and control-populations will be excluded as the algorithm proceeds, unless some of those SNPs are involved in a pattern of differential-expression specific to the remaining case-subjects, in which case they will be retained (as desired). Third, the method also allows us to correct for

continuous covariates. This covariate-correction is described in detail in [64], but essentially amounts to a reweighting of the $Q(j, k)$ in Step-1 to reduce the overall level of differential-expression contributed by structures which are not evenly distributed in covariate-space. Finally, the method itself is rather straightforward and does not require the fine-tuning of parameters.

As mentioned in Step-2, the overall level of differential-expression between $D(i)$ and $X$ at each iteration is recorded as the trace $\bar{Q}(i)$. The significance-level of $\bar{Q}(\cdot)$ is determined with respect to a null hypothesis (H0) which assumes that the heterogeneity is independent of case- and control-labels. Samples from H0 are drawn by randomly permuting the case- and control-labels in arm-1 (i.e., randomly interchanging rows of $D$ and $X$) while respecting proximity in covariate-space. By comparing the values of the $\bar{Q}(i)$ from the original data to the distribution of $\bar{Q}(i)$ associated with the null-hypothesis, we assign an (empirical) training-$p$-value to the individual $\bar{Q}(i)$ for each iteration $i$. Similarly, we calculate an overall empirical training-$p$-value (across all iterations), which estimates the probability that the trace $\bar{Q}$ from the original data-set could be drawn from the null-hypothesis.

Within this context, the detection of a disease-specific bicluster corresponds to an elevated (i.e., statistically-significant) value of $\bar{Q}(i)$. The case-subjects and allele-combinations comprising the bicluster can then be approximated by the subsets $\mathcal{J}(i)$ and $\mathcal{K}(i)$ for those $i$.

## Replication

When discussing any particular replication-arm (e.g., arm-2), we will use primed indices (e.g., cases and controls will be indexed via $j'_D$ and $j'_X$). To assess replication we first consider the set of allele-combinations $\mathcal{K}'$ available within the replication-arm. This subset will limit the alleles we can use from within the original training-arm (i.e., arm-1). For any iteration $i$, we select the allele-subset $\mathcal{K}(i)$ from the training-data-set, and then construct the intersection $\mathcal{K}'(i) := \mathcal{K}(i) \cap \mathcal{K}'$. For the replication-arm arm-2 the allele set $\mathcal{K}'(i)$ will have a size $N'(i)$, which is typically around 85% of $N(i)$ (i.e., 85% of the full size of $\mathcal{K}(i)$). For the other replication-arms (i.e., arms 3 and 4) the overlap will be lower. Using $\mathcal{K}'(i)$ as well as the case-subject subset $\mathcal{J}(i)$, we define the $M(i) \times N'(i)$ submatrix $D'(i)$ within the training data (note $D'(i)$ is a submatrix of the $M(i) \times N(i)$ submatrix $D(i)$ defined above). We then calculate the dominant SNP-wise principal-component $v(i) \in \mathbb{R}^{N'(i)}$ of $D'(i)$.

We project each subject within the training-data-set onto $v(i)$, producing a 'bicluster-score' (i.e., a single number) $u_{j_D}(i)$ for each case-subject in the training-data-set, and $u_{j_X}(i)$ for each control-subject in the training-data-set (recall that $j_D$ and $j_X$ index the case- and control-subjects in the training-data-set). Based on the definition of the bicluster, we expect that the typical values of $u_{j_D}(i)$ will be larger than the typical values of $u_{j_X}(i)$. We measure this difference by calculating the area under the receiver-operator-characteristic curve (**A U C**) between the sets $\{u_{j_D}(i)\}$ and $\{u_{j_X}(i)\}$; we refer to this **A U C** as $A(i)$. When calculating $A(i)$ we correct for the same ancestry-related covariates as in [20] (see Methods and [60]).

We also project each subject in the replication-arm onto the same vector $v(i)$, producing bicluster-scores $u'_{j'_D}(i)$ for each case-subject in the replication-arm, and $u'_{j'_X}(i)$ for each control-subject in the replication-arm. Once again, we expect that the typical values of $u'_{j'_D}(i)$ will be larger than the typical values of $u'_{j'_X}(i)$ in the replication-arm. We measure this difference by calculating the **A U C** $A'(i)$, once again correcting for the ancestry-related covariates.

We assess the overall significance of the replication by considering a null-hypothesis where the structure of the replication-arm is independent of disease-status. We can draw a sample from this null-hypothesis (H0') by randomly permuting the case- and control-labels within the

replication-arm (while respecting proximity in covariate-space). In this manner we compare the original replication **A U C** $A'(\cdot)$ (as a function of $i$) to the distribution of $A'(\cdot)$ obtained under H0'.

Later on below (e.g., Fig 3) we calculate the average $\bar{A}'$ of $A'(\cdot)$ over a range of iterations, and then compare $\bar{A}'$ to the distribution of $\bar{A}'$ obtained under this label-shuffled null-hypothesis. We define the range of iterations by taking an interval which is significant for both the trace $\bar{Q}(i)$ and the **A U C** $A(i)$ defined using only the training-arm. For example, in Fig 3 we consider the range of iterations $i \in [175, 350]$.

## Polygenic-Risk-Scores (PRSs)

We calculate PRSs using the general strategy from [20], and further described in page 60 of the S1 Text within that paper. To briefly summarize: We use the genotype-level data from [20], which was imputed using the 1KG reference-panel. We then run a GWAS on this genotype-level data. This GWAS produces summary-statistics defined by contrasting cases and controls from the training-arm, while correcting for ancestry-related covariates. Once we have the summary-statistics defined by the GWAS, we run Plink's 'clump' function to account for LD. We perform this clumping step using the same parameters as in [20] (e.g., info-score threshold of 0.9, $R^2$-threshold of 0.1, genomic window of 500Kb, and minor-allele-frequency threshold of 0.05.) As a technical note: our ultimate goal is to analyze these PRS scores in the context of our heterogeneity analysis, which can be influenced by subtle relationships between SNPs. Consequently, we wanted to use the most accurate available information regarding LD. After the initial data-sets described in [20] were published, the Haplotype Reference Consortium European Reference Panel (HRC EUR panel) became available through the Wellcome Trust Sanger Institute [69]. This HRC EUR panel dramatically increased the amount of information available for approximating LD, and we use this panel when clumping our summary statistics. Finally, after clumping, we use the assigned weights for each SNP to form a PRS. We test the performance of this PRS on our replication-arms.

For any subject $j'$ within a particular replication-arm, we denote by $\mathbf{PRS}_{\text{wide}}(j')$ the 'population-wide' PRS defined by contrasting *all* the cases in the training-arm with the controls in the training-arm (when generating the summary-statistics). We further denote by $\mathbf{PRS}_{\text{wide}}(j'; \tilde{p})$ the population-wide PRS constructed after restricting the SNP-weight-vector to include only those SNPs with individual GWAS $p$-values that are more significant than the threshold $\tilde{p}$ (when forming the PRS).

We also define a 'bicluster-informed' PRS, denoted by $\mathbf{PRS}_{\text{bicl}}(j'; i)$, by contrasting *only* the cases in $D(i)$ with the controls from the training-arm (when generating the summary-statistics). We further denote by $\mathbf{PRS}_{\text{bicl}}(j'; i, \tilde{p})$ the bicluster-informed PRS constructed after restricting the SNP-weight-vector to include only those SNPs with individual GWAS $p$-values that are more significant than the threshold $\tilde{p}$ (when forming the PRS). With this notation $\mathbf{PRS}_{\text{wide}}(j')$ and $\mathbf{PRS}_{\text{wide}}(j'; \tilde{p})$ are equivalent to $\mathbf{PRS}_{\text{bicl}}(j'; 1)$ and $\mathbf{PRS}_{\text{bicl}}(j'; 1, \tilde{p})$, respectively. However, we will typically consider $\mathbf{PRS}_{\text{bicl}}$ for iterations $i \in [175, 350]$; in this range $\mathbf{PRS}_{\text{wide}}(j'; \tilde{p})$ and $\mathbf{PRS}_{\text{bicl}}(j'; i, \tilde{p})$ will differ.

We measure the performance of the population-wide $\mathbf{PRS}_{\text{wide}}(j')$ by calculating the **AUC**-wide between the case-values $\{\mathbf{PRS}_{\text{wide}}(j'_D)\}$ and the control-values $\{\mathbf{PRS}_{\text{wide}}(j'_X)\}$, once again correcting for the ancestry-related covariates. Similarly, we measure the performance of $\mathbf{PRS}_{\text{wide}}(j'; \tilde{p})$, $\mathbf{PRS}_{\text{bicl}}(j'; i)$ and $\mathbf{PRS}_{\text{bicl}}(j'; i, \tilde{p})$ by calculating the associated **AUCs**, denoted by $\text{AUC}_{\text{wide}}(\tilde{p})$, $\mathbf{AUC}_{\text{bicl}}(i)$ and $\text{AUC}_{\text{bicl}}(i, \tilde{p})$, respectively.

### Gene-enrichment analysis

We perform a simple over-representation analysis using the `go_bp` ontology from Seek [70]. We restrict our attention to the 132 neuronally-related pathways (i.e., those referencing neurons, synapses or axons). For any given iteration $i$ we consider the remaining allele-combinations within $\mathcal{K}(i)$, retaining those genes which have more than half their originally associated alleles remaining. These retained genes form a gene-set $\mathcal{G}(i)$ which we then overlap with each pathway $\mathcal{H}_l$ to obtain the intersection $\mathcal{G}(i) \cap \mathcal{H}_l$. From this intersection we obtain the gene-count $\kappa(i, l) = |\mathcal{G}(i) \cap \mathcal{H}_l|$ for pathway $l$ at iteration $i$.

We assess the significance of the gene-counts by considering the same null-hypothesis H0 used when biclustering. We compare each of the $\kappa(i, l)$ to the distribution of $\kappa(i, l)$ obtained under the label-shuffled null-hypothesis. Later on below we calculate the average z-score $\bar{z}$ of the $\kappa(i, l)$ over a range of iterations and all the neuronally-related pathways, and then compare that $\bar{z}$ to the distribution of $\bar{z}$ obtained under H0.

## Results

We apply the half-loop-counting algorithm (see Methods) to the 'BDRN' cohort used as the training arm. The trace $\bar{Q}(\cdot)$ associated with the original data is shown in red in Fig 2. Were the signal homogeneous, we would expect to see a trace that starts out high and gradually decreases in magnitude. Instead, we see a trace that behaves non-monotonically, and is statistically insignificant for a range of iterations. The trace from the original data (in red) attains values that are significantly higher than the majority of the traces one would expect under the null-hypothesis (black) near iteration $i \sim 175$. This is an indicator that the data is heterogeneous, and that a bicluster has been detected near iteration $i \sim 175$; the identity of the bicluster can be approximated by one of the submatrices $D(i)$ where the training-$p$-value is large. We can calculate the empirical $p$-value associated with the entire trace $\bar{Q}(\cdot)$ by comparing the red curve (across all iterations) to the black curves, estimating an overall p-value of $p \lesssim 1/64$.

In idealized scenarios where the 'true' bicluster is sharply defined, the trace typically has a sharp peak near the $D(i)$ that most closely corresponds to the bicluster [63, 64]. However, in this case while the trace has a peak at around $i \sim 175$, this peak is not particularly sharp, and the trace is nearly as significant across a range of iterations $i \in [175, 350]$. The largest of these submatrices (i.e., $D(175)$) corresponds to $\sim 47\%$ of the case-subjects and $\sim 31\%$ of the allele-combinations. The smallest of these submatrices (i.e., $D(350)$) corresponds to $\sim 21\%$ of the case-subjects and $\sim 9\%$ of the allele-combinations.

This 'plateau' of significance indicates that the true signal is not a perfectly crisp and well-delineated bicluster. Instead, this plateau suggests that, while there are certain 'core' case-subjects that exhibit a strong similarity across certain allele-combinations, there are additional case-subjects that are 'adjacent' to those in the core. These adjacent subjects exhibit a slightly weaker similarity involving a slightly expanded set of allele-combinations. Consequently, we expect iterations in the interval $i \in [175, 350]$ to provide a range of approximations to the true 'core' signal (which is still unknown). One could certainly select the iteration with the highest training-$p$-value to approximate the bicluster, but as nearby iterations have nearly the same training-$p$-value, we expect them to also provide reasonable estimates of the true signal.

Given our approximation to the signal described above from the training-data-set, we test for replication in each of the replication-arms 2, 3 and 4. We are interested in how strongly our approximate signal replicates, as well as whether our approximation has been compromised by overfitting. Because the signal spans a range of iterations in arm-1, we assess the extent of replication across the plateau $i \in [175, 350]$. This interval corresponds to significant values of the trace $\bar{Q}(i)$ as well as the **AUC** $A(i)$ defined only using the training-data.

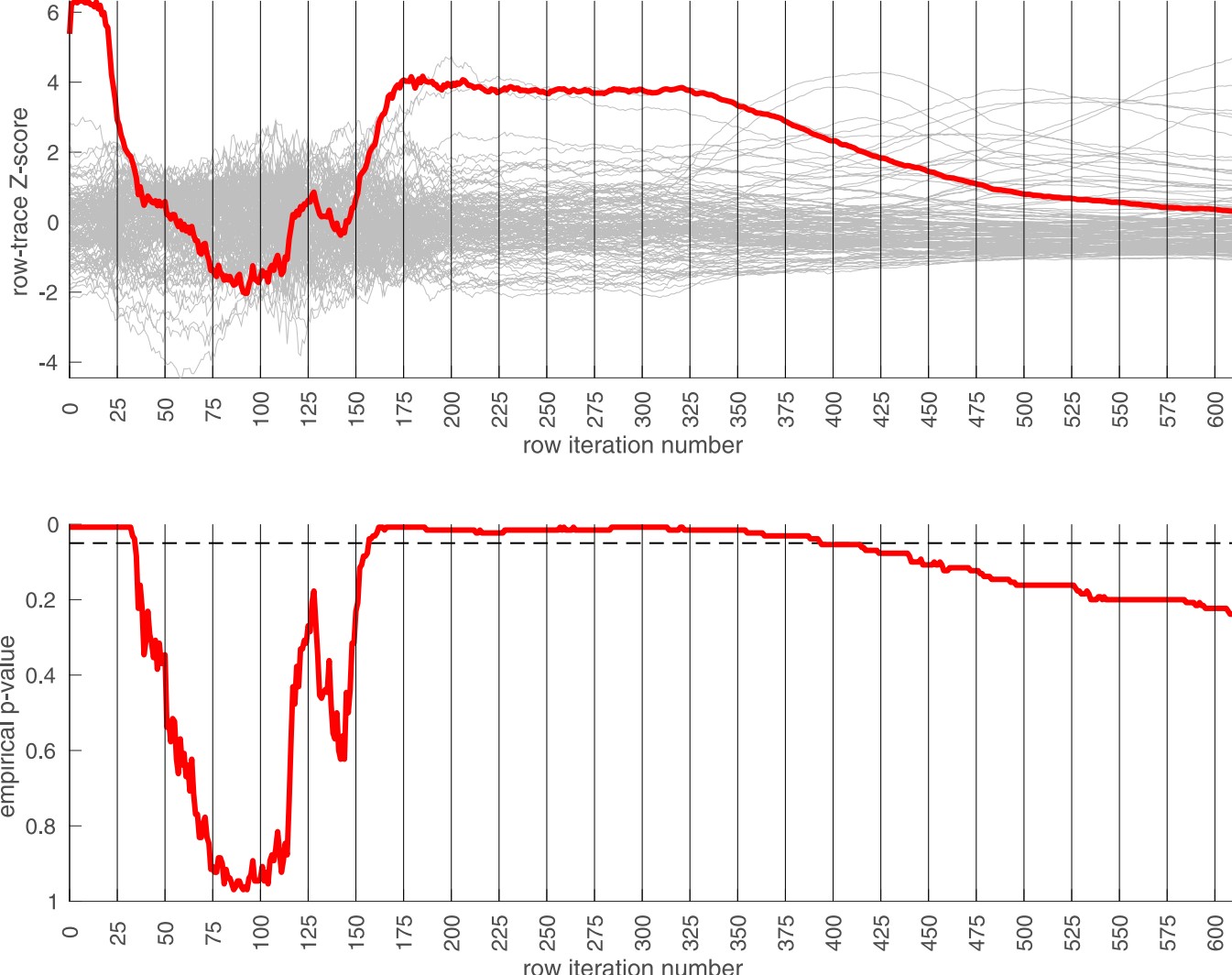

**Fig 2. In this figure we show the output of the half-loop biclustering algorithm applied to the BDRN cohort in arm-1 (limited to those SNPs with maf ≥0.25).** As described in the main text, the algorithm proceeds iteratively, eliminating rows and columns from the case-subject-array $D$ until all have been removed. At each iteration $i$, the remaining submatrix $D(i)$ comprises case-subjects $\mathcal{J}(i)$ and allele-combinations $\mathcal{K}(i)$. At each iteration we record the 'row-trace' $\bar{Q}(i)$, which is the covariate-corrected average level of differential-expression between $D(i)$ and the control-subjects $X$. In the top row of subplots we show the row-trace for the data (red) as well as for 128 label-shuffled trials (black). Each of the row-traces has been transformed into an iteration-dependent z-score (estimated using the distribution of label-shuffled trials at that iteration). In the bottom row we show the corresponding empirical p-value, as estimated for each iteration using the label-shuffled trials. The dashed black-line corresponds to the 95th percentile (i.e., a significance value of 0.05 if each iteration were considered independently). If the signal were homogeneous we would expect to see the red trace begin at a high value and decay relatively monotonically. By contrast, we see strong evidence for heterogeneity; the red trace is far from monotonic. The overall p-value for the data (red-trace), estimated using the strategy in [64], is $p \lesssim 1/64$. Note that the trace is significant over a range of iterations, including $i \in [175, 350]$.

The results of this replication study for arm-2 are shown in Fig 3. The top subplot illustrates the **AUC** $A(i)$ (red) and $A'(i)$ (green) as a function of $i$. The bottom subplot shows the associated $p$-value for each $i$ (under a label-shuffled null-hypothesis). Note that the training-**AUC** $A(i)$ is high over the range of iterations $i \in [175, 350]$ for which the training-$p$ value is significant. Note also that the peak of $A(i)$ occurs within a few iterations of the peak of the training $p$-value. This correspondence corroborates the claims made above: we believe we have detected a disease-related signal within the training-data-set that involves only a subset of subjects and

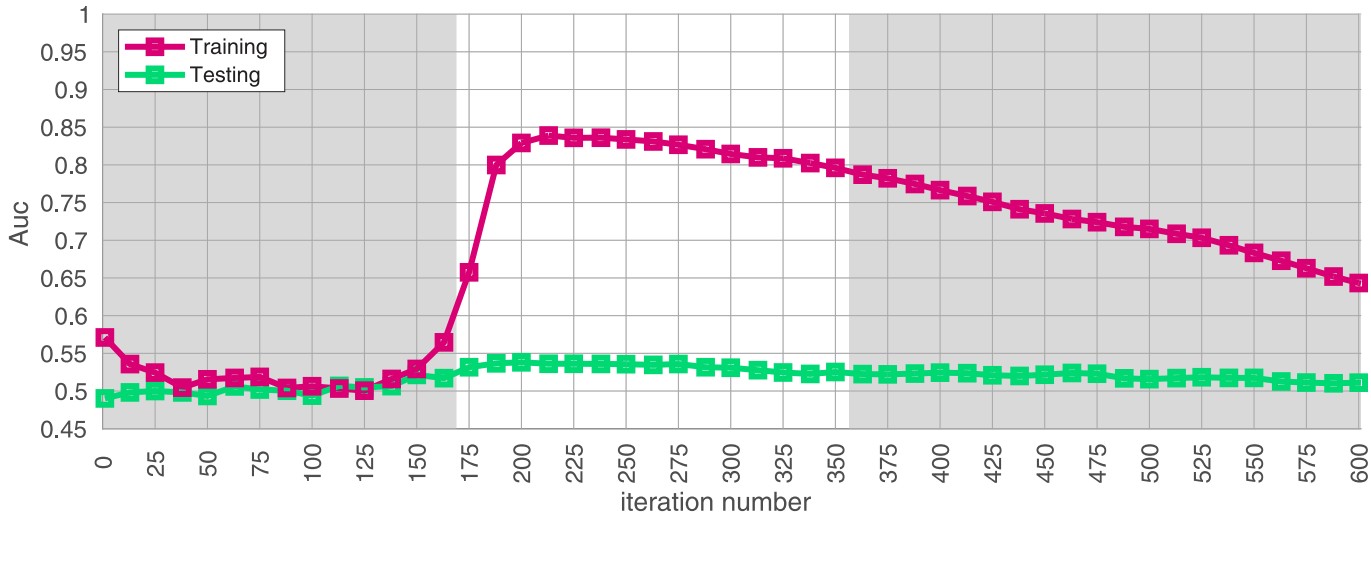

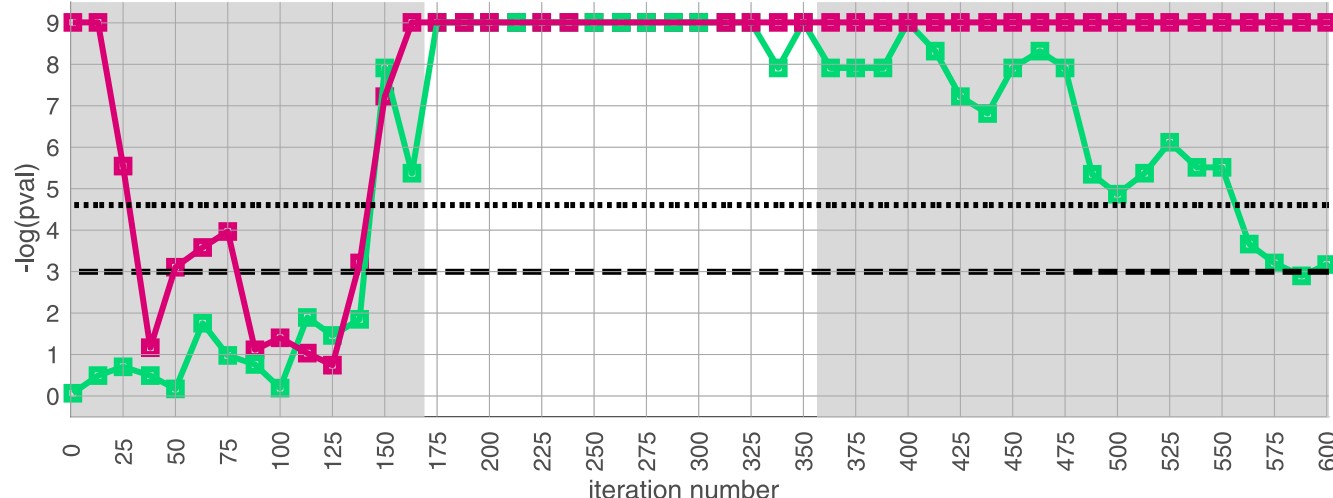

**Fig 3. In this figure we illustrate the replication of the bicluster in arm-2.** Note that the SNP-overlap between arm-1 and arm-2 is $\sim$ 85%. On the top we show $A(i)$ in red and $A'(i)$ in green. On the bottom we show the associated p-values for $A(i)$ and $A'(i)$, calculated with respect to $H0$ and $H0'$ for each iteration individually. Standard significance-levels 0.05 and 0.01 are shown in dashed- and dotted-lines, respectively. The interval $i \in [175, 350]$ is highlighted in white. Note that both $A(i)$ and $A'(i)$ have peaks within the range that the trace was significant (c.f. Fig 2). The overall replication for arm-2 within the interval $i \in [175, 350]$ is estimated at $p \lesssim 10^{-12}$.

alleles. While the magnitude of the replication-**AUC** $A'(i)$ is lower than the training-**AUC** $A(i)$, the value of $A'(i)$ is also statistically significant over the range of iterations $i \in [175, 350]$, with a peak at roughly the same point.

Similar results for arm-3 and arm-4 are shown in Figs 4 and 5. Note that the SNP-overlap between these arms and the training-data-set is quite a bit lower than that for arm-2. Recall that arm-2 has a overlap of $\sim$ 85% with the SNPs in arm-1, while arm-3 and arm-4 have overlaps of $\sim$ 50% and $\sim$ 30%, respectively.

We believe that this reduction in SNP-overlap is partially responsible for the reduction in the magnitude of replication-**AUC**s observed in these arms. To test this hypothesis, we randomly eliminate SNPs from arm-2 until the SNP-overlap between the training-data-set and

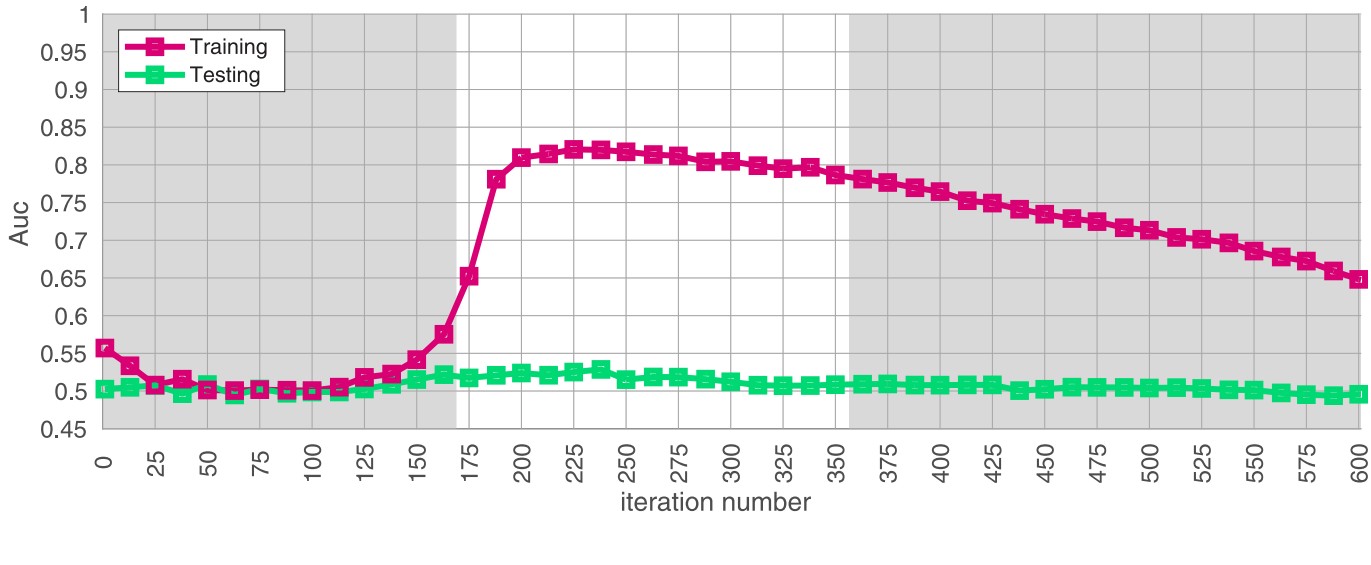

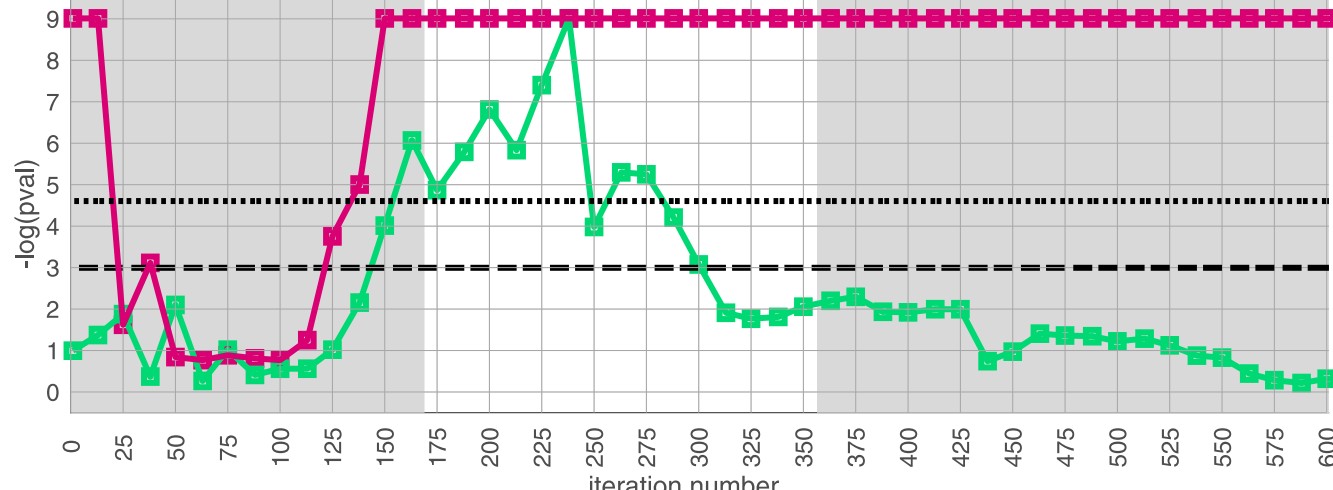

**Fig 4. This figure is similar to Fig 3, except that we use arm-3 instead of arm-2.** The overall replication for arm-3 within the interval $i \in [175, 350]$ is estimated at $p \lesssim 10^3$. Note that the SNP-overlap between arm-1 and arm-3 is only $\sim 50\%$.

arm-2 is equal to the SNP-overlap between the training-data-set and arm-3. The results of this replication-study are shown in Fig 13 in S1 Text: note that the amplitude of $A'(i)$ has degraded in comparison to the values shown in Fig 3. We then randomly eliminate even more SNPs, until the SNP-overlap between the training-data-set and arm-1 is equal to the SNP-overlap between the training-data-set and arm-4 (see Fig 14 in S1 Text), and the amplitude $A'(i)$ degrades even further. More generally, by reducing the number of SNPs we include in the replication-arm, we can cause the values of $A'(i)$ to drop; depending on the subset of SNPs retained, the values of $A'(i)$ for arm-2 can be reduced to values similar to those observed in arm-3 and arm-4.

In summary, the **AUC** associated with the genotype-based bicluster score discovered in the training-data-set replicates to varying degrees across all 3 replication arms. In each case the average $A'(i)$ calculated over the interval $i \in [175, 350]$ was significantly larger than what one would expect were the case- and control-labels in the replication-arm randomly permuted

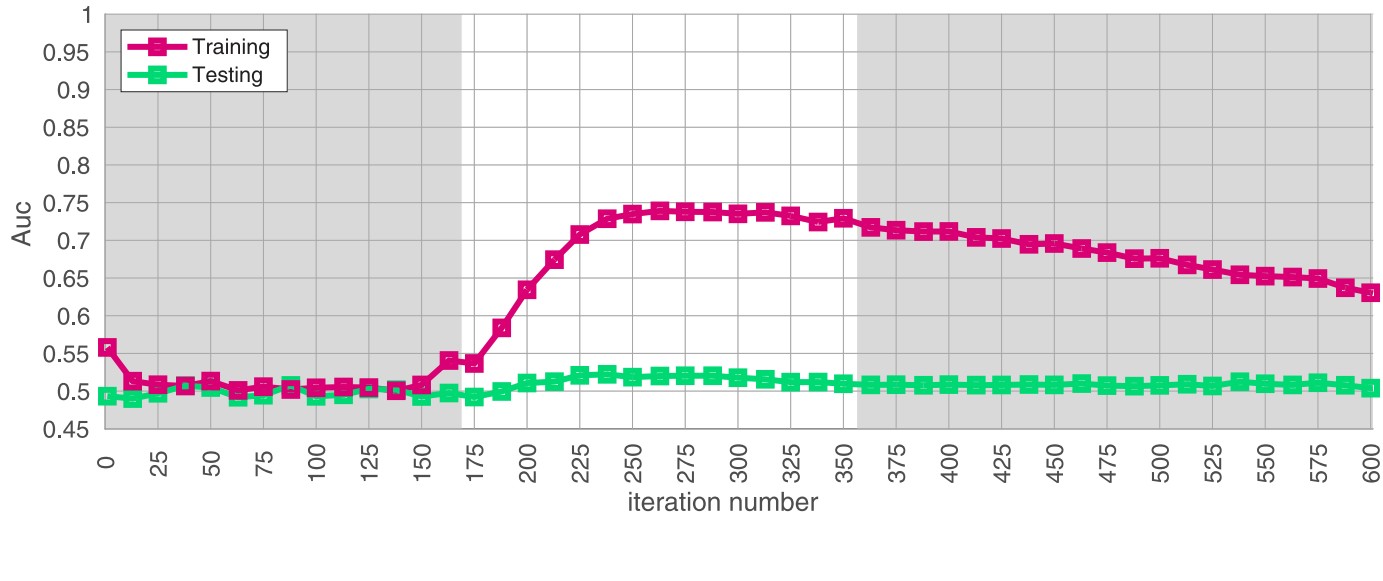

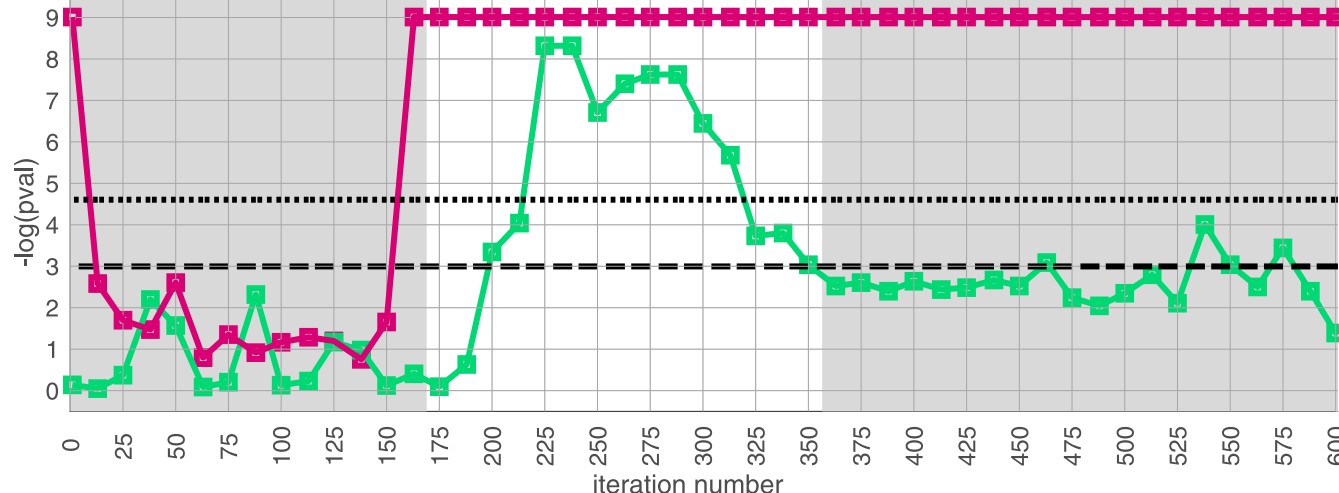

**Fig 5. This figure is similar to Fig 3, except that we use arm-4 instead of arm-2.** The overall replication for arm-3 within the interval $i \in [175, 350]$ is estimated at $p \lesssim 10^3$. Note that the SNP-overlap between arm-1 and arm-4 is only $\sim 30\%$.

($p \lesssim 1/1000$). Consequently, we are fairly certain that—while our approximation of the bicluster is far from perfect—we have indeed identified a robust disease-related signal which generalizes across a variety of different BD studies.

## Discussion

### Interaction with covariates

Given the observations above, it is natural to ask what might be driving the signal associated with this bicluster. We first checked to see if the bicluster was driven by the ancestry-related covariates in our data-set. As shown in Figs 15 and 16 in S1 Text, the subjects in the bicluster have a distribution of ancestries similar to the remainder of arm-1 (recall that we corrected for ancestry as a covariate). By considering the subjects remaining in $D(i)$, we also determined that the bicluster does not seem to be associated with sex (see Fig 17 in S1 Text).

## Interaction with BD subtype

We then checked to see if the bicluster was associated with bipolar subtype. We measured the fraction of subjects classified as bipolar-type-1 versus bipolar-type-2 as our algorithm proceeded. Specifically, we measured the fraction of case-subjects in $\mathcal{J}(i)$ that were classified as BDI and BDII. If the bicluster were driven by BDII subjects, then we would expect the proportion of remaining BDII case-subjects to increase with the iteration-index $i$. Conversely, if the bicluster were driven by BDI subjects, then we would expect the proportion of remaining BDI case-subjects to increase with iteration-index. As shown in Fig 6, we found that this latter scenario holds; the bicluster was significantly enriched for BDI relative to BDII. This enrichment for BDI also impacts our risk-prediction results (see below). Note that, when determining this enrichment, we compare the proportion of BDI and BDII case-subjects at each iteration to the proportion at iteration $i = 1$ (i.e., across all case-subjects in arm-1). In this manner our enrichment is defined relative to the starting proportion of BDI and BDII subjects in our training-arm, and is not influenced by the recruitment rates for BDI and BDII (which can differ across studies).

While significant, this BDI-enrichment was not completely overwhelming: the initial fraction of BDII participants in arm-1 was $\sim 31\%$, which dropped to $\sim 26\%$ at iteration $i = 240$. Thus, while the majority of the case-subjects in the bicluster are classified as BDI, those classified with BDII do still contribute to the overall signal. It is possible that this BDI-enrichment is due to a true difference between the BD-subtypes at the genetic level. However, it is also possible that this enrichment is partially driven by inaccuracies associated with classification [14].

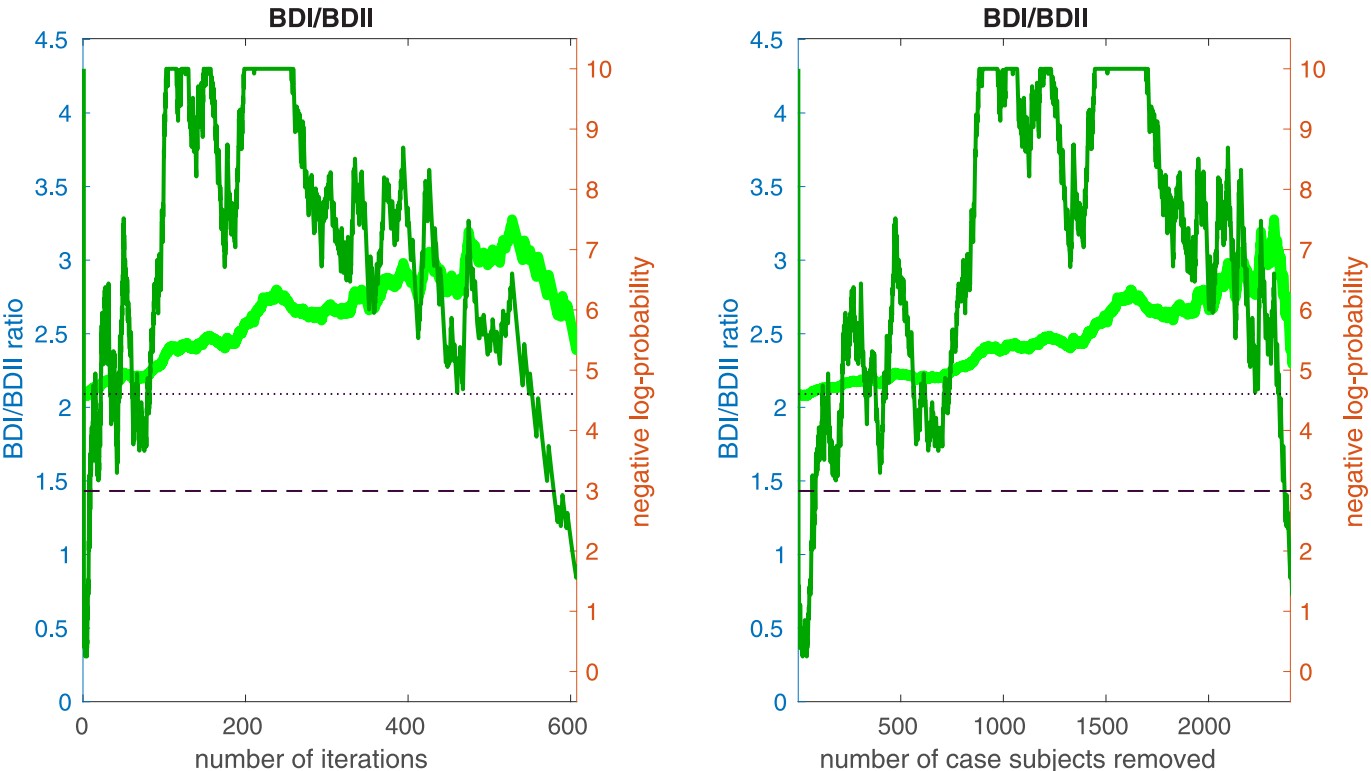

**Fig 6.** This figure plots the ratio of BDI to BDII subjects within $\mathcal{J}(i)$ (light-green, left y-axis) as a function of the iteration $i$ (left) and the number of removed case-subjects (right). The dark-green line corresponds to the negative-log-probability (right y-axis) of observing a ratio at least as large by chance. The dashed and dotted horizontal lines indicate 0.05 and 0.01 significance values, respectively. Note that the BDI population is over-represented across a range of iterations including $i \in [175, 350]$, implying that the bicluster we observe is significantly enriched for BDI subjects.

## Bicluster-informed PRS performance

As described in the Methods section, we calculated the population-wide $\mathbf{PRS}_{\text{wide}}(j'; \tilde{p})$ and the bicluster-informed $\mathbf{PRS}_{\text{bicl}}(j'; i, \tilde{p})$ across a variety of iterations $i$ and $\tilde{p}$-thresholds. We compared the bicluster-informed $\mathbf{PRS}_{\text{bicl}}(j'; i, \tilde{p})$ performance to the one generated by the population-wide $\mathbf{PRS}_{\text{wide}}(j'; \tilde{p})$ across a variety of $\tilde{p}$-thresholds. Results for arm-2 are shown in Fig 7. Results for arm-3 and arm-4 are shown alongside arm-2 in Fig 8, and individually in Figs 23 and 24 in S1 Text.

Note that, when constructing $\mathbf{PRS}_{\text{bicl}}(j'; i, \tilde{p})$, we restrict ourselves to a subset of case-subjects within the training-arm determined by $\mathcal{J}(i)$. In this case, when $i \in [175, 350]$ the case-subset $\mathcal{J}(i)$ retains only $\sim 50\% - 20\%$ of the original case-subjects in arm-1. Typically, one might expect a reduction in the number of case-subjects to yield a corresponding reduction in power, giving rise to a reduced discriminability in the testing-arms 2,3 and 4. However, as we see in Fig 8, the discriminability for $\mathbf{PRS}_{\text{bicl}}(j'; i, \tilde{p})$ is typically *higher* than $\mathbf{PRS}_{\text{wide}}(j', \tilde{p})$ when $i \in [175, 350]$. This suggests that the case-subjects in $\mathcal{J}(i)$ identified by the bicluster correspond to a stronger genetic signal, likely arising from the increased homogeneity within $\mathcal{J}(i)$.

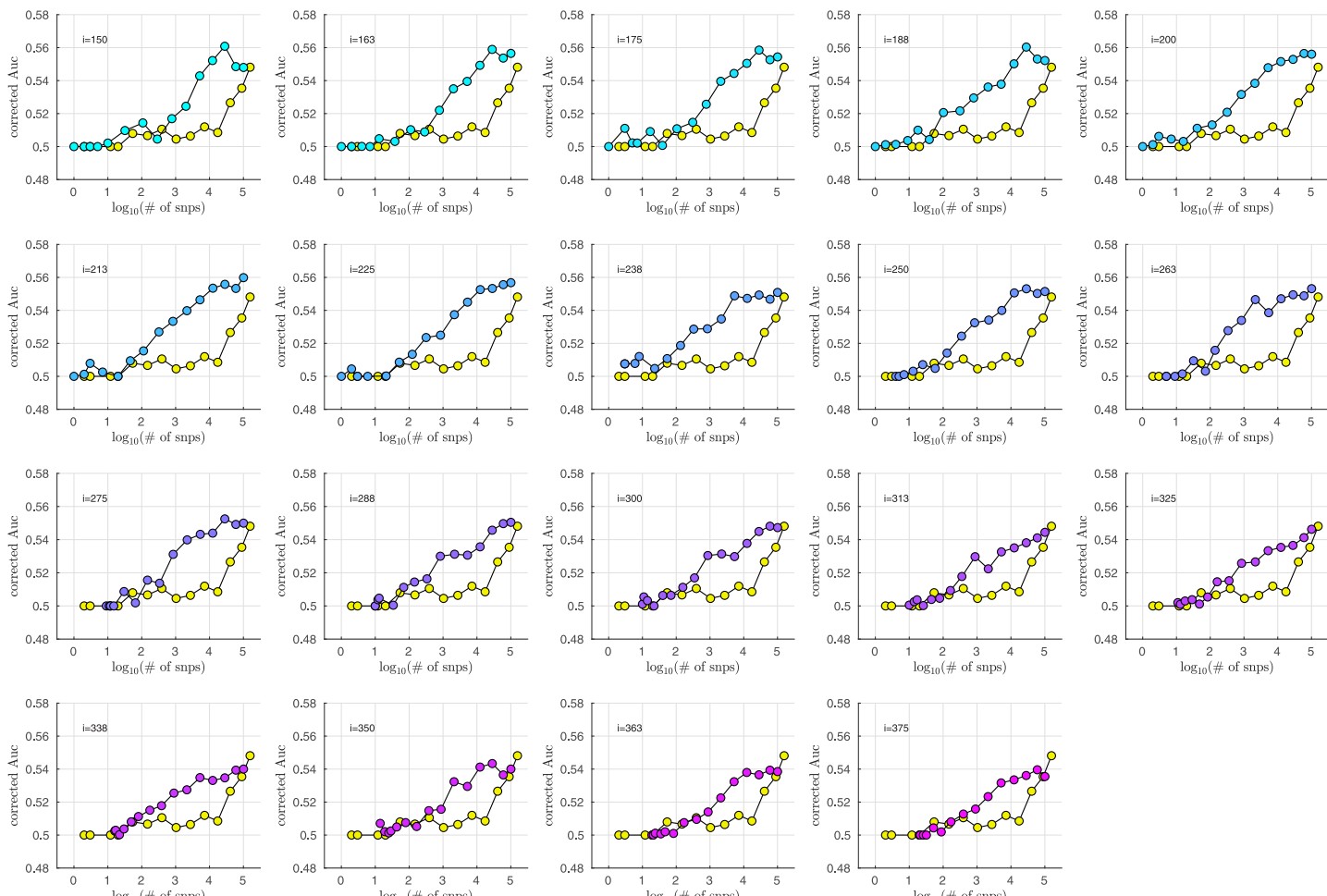

**Fig 7. In each subplot we show in yellow the $\mathbf{AUC}_{\text{wide}}(\tilde{p})$ (vertical) for arm-2 as a function of the number of SNPs corresponding to each $\tilde{p}$-threshold (horizontal, log-scale).** Additionally, we show $\mathbf{AUC}_{\text{bicl}}(i, \tilde{p})$ for a particular iteration $i$ (with $i$ varying across subplots). The color-code used for $\mathbf{AUC}_{\text{bicl}}(i, \tilde{p})$ ranges from blue to pink, corresponding to the iteration index $i$. Note that, by using the bicluster to inform the PRS, the performance typically improves. This improvement in performance becomes marked when the number of SNPs is limited to a relatively small fraction of the total (e.g., $\sim 1\%$ of the total, corresponding to a $\log_{10}(\#)$ of $\sim 3$).

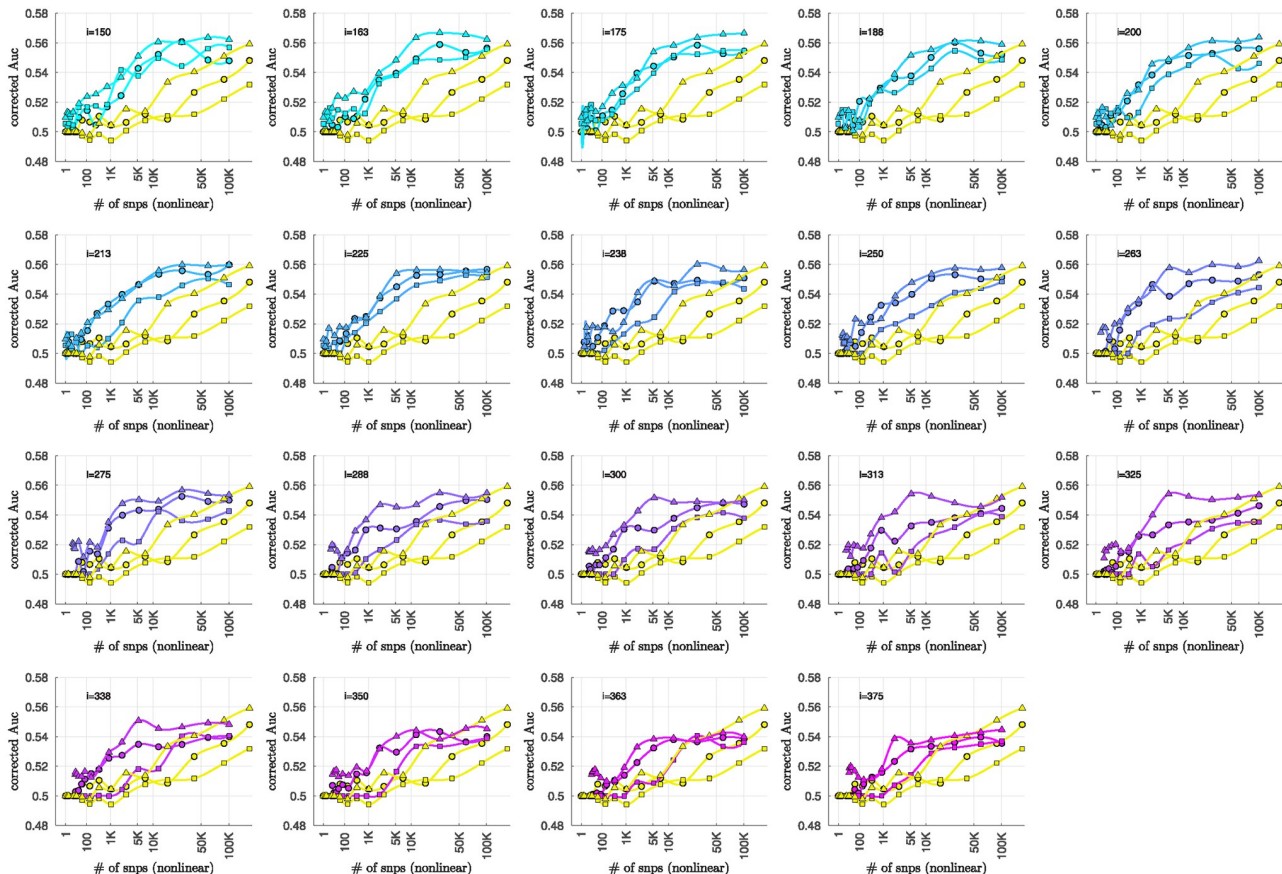

**Fig 8. This figure uses circles to displays the same information as Fig 7 (corresponding to replication arm-2).** In this figure we use an algebraic-scale for the horizontal-axis (rather than a log-scale) in order to better emphasize the interval where the number of SNPs used is between $1K$ and $10K$. The results for replication arm-3 and arm-4 are shown using squares and triangles, respectively.

Note that $\mathbf{PRS}_{\text{bicl}}$ and $\mathbf{PRS}_{\text{wide}}$ are not capturing identical signals (see the Nagelkerke $R^2$ analysis in the S1 Text). It is useful to compare the performance of $\mathbf{PRS}_{\text{bicl}}$ with $\mathbf{PRS}_{\text{wide}}$ as there are features of $\mathbf{PRS}_{\text{bicl}}$ which indicate that it is more robust than $\mathbf{PRS}_{\text{wide}}$. As one example, we point out that $\mathbf{AUC}_{\text{bicl}}(i, \tilde{p})$ is markedly higher than $\mathbf{AUC}_{\text{wide}}(\tilde{p})$ when the number of SNPs used (denoted by $N_{\text{SNP}}$) is fewer; one begins to see the effect between $1K$ and $10K$. This suggests that the bicluster-informed $\mathbf{PRS}_{\text{bicl}}(j'; i, \tilde{p})$ is not only outperforming the population-wide $\mathbf{PRS}_{\text{wide}}(j'; \tilde{p})$, but also correctly attributing the largest PRS-weights to those SNPs that truly carry the signal (and which are most important for replication). As one illustration, by comparing the values of $\mathbf{AUC}_{\text{bicl}}$ to $\mathbf{AUC}_{\text{wide}}$ in Fig 8, we can directly see that the bicluster-informed PRS would replicate across arms 2,3 and 4 for values of $i = 225$ and $N_{\text{SNP}} \in [10^3, 10^4]$, while the population-wide PRS would not.

Motivated by the significant BDI-enrichment seen within the training-arm (see Fig 6), we repeated these assessments for the BDI- and BDII-populations within the testing-arms. More specifically, recall that, for any particular testing-arm, the $\mathbf{AUC}_{\text{wide}}(\tilde{p})$ and $\mathbf{AUC}_{\text{bicl}}(i, \tilde{p})$ values shown in Figs 7 and 8 are defined using the values of $\mathbf{PRS}_{\text{bicl}}(j'; i, \tilde{p})$ across all case- and control-subjects $j'$ for that testing-arm. We can now use the same values of $\mathbf{PRS}_{\text{wide}}(j'; \tilde{p})$ and $\mathbf{PRS}_{\text{bicl}}(j'; i, \tilde{p})$, but only compare the BDI-case-subjects to the control-subjects in the testing-arm. This produces 'restricted' $\mathbf{AUC}$-values, which we denote by $\mathbf{AUC}_{\text{wide}|\text{BDI}}(\tilde{p})$ and

$\mathbf{AUC}_{\mathrm{bicl|BDI}}(i, \tilde{p})$, respectively. In a similar fashion we can restrict the case-subjects in the testing-arm to the BDII-case-subjects, and calculate $\mathbf{AUC}_{\mathrm{wide|BDII}}(\tilde{p})$ and $\mathbf{AUC}_{\mathrm{bicl|BDII}}(i, \tilde{p})$.

The results are shown in Figs 9 and 10, respectively. Note that the improvement to risk-prediction persists for the BDI-population, but is not as robust for the BDII-population. The performance of $\mathbf{AUC}_{\mathrm{bicl|BDII}}(i, \tilde{p})$ is particularly poor for the BDII-population in arm-3, for which there were only $M = 435$ BDII-subjects (i.e., the fewest out of all the arms). It is possible that the variation in the performance of $\mathbf{AUC}_{\mathrm{bicl|BDII}}(i, \tilde{p})$ for the BDII-population across the replication-arms has to do with these differences in power. It is also possible that there are other systematic issues affecting the BDII-population, including variation in the life history of the subjects or the metrics used for their clinical diagnosis [14].

To summarize the overall relationship between BD-subtype, the bicluster-informed PRS and the population-wide PRS, we pool the subjects across the replication-arms and convert the combined $\mathbf{AUC}$-values into $R^2$-values on a liability-scale [71] using prevalences of 2% for BD, and 1% for BDI and BDII [72]. Using notation analogous to the $\mathbf{AUC}$-values, we denote these liability-scores as $\mathcal{R}^2_{\mathrm{wide}}(\tilde{p})$, $\mathcal{R}^2_{\mathrm{wide|BDI}}(\tilde{p})$ and $\mathcal{R}^2_{\mathrm{wide|BDII}}(\tilde{p})$, as well as $\mathcal{R}^2_{\mathrm{bicl}}(i, \tilde{p})$, $\mathcal{R}^2_{\mathrm{bicl|BDI}}(i, \tilde{p})$ and $\mathcal{R}^2_{\mathrm{bicl|BDII}}(i, \tilde{p})$, respectively. The resulting liability-scores are shown in Fig 11.

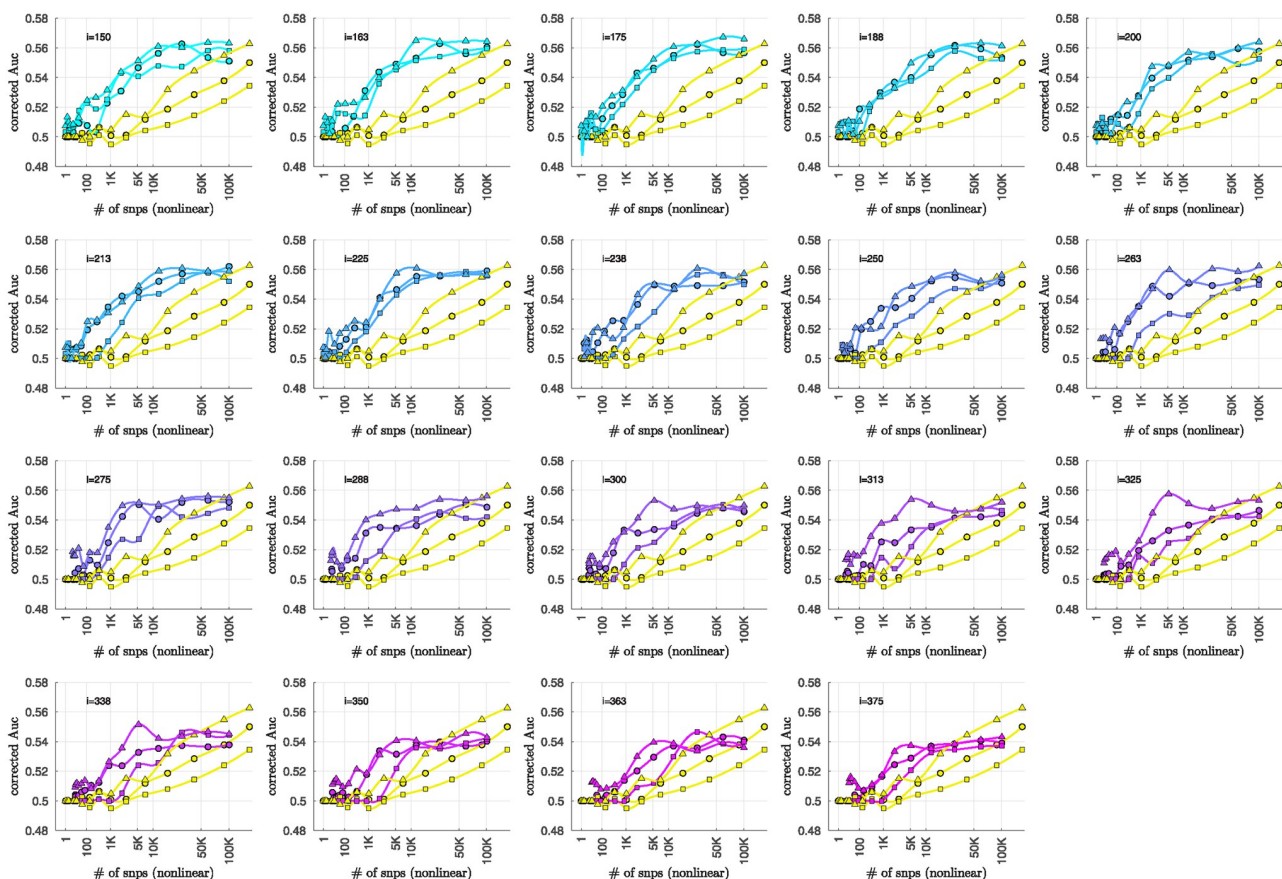

**Fig 9. This figure is similar to Fig 8, except that we limit ourselves only to those case-subjects in the replication-arms which are classified as BDI.** This subset corresponded to 66% ($M = 3834$), 84% ($M = 2995$) and 75% (M = 5107) of the case-population for arms 2, 3 and 4, respectively. The corresponding **AUC**-values are denoted by $\mathbf{AUC}_{\mathrm{wide|BDI}}(\tilde{p})$ and $\mathbf{AUC}_{\mathrm{bicl|BDI}}(\tilde{p})$ in the main text. For reference the training-arm had $M = 1645$ BDI case-subjects, corresponding to 65% of the case-population in arm-1.

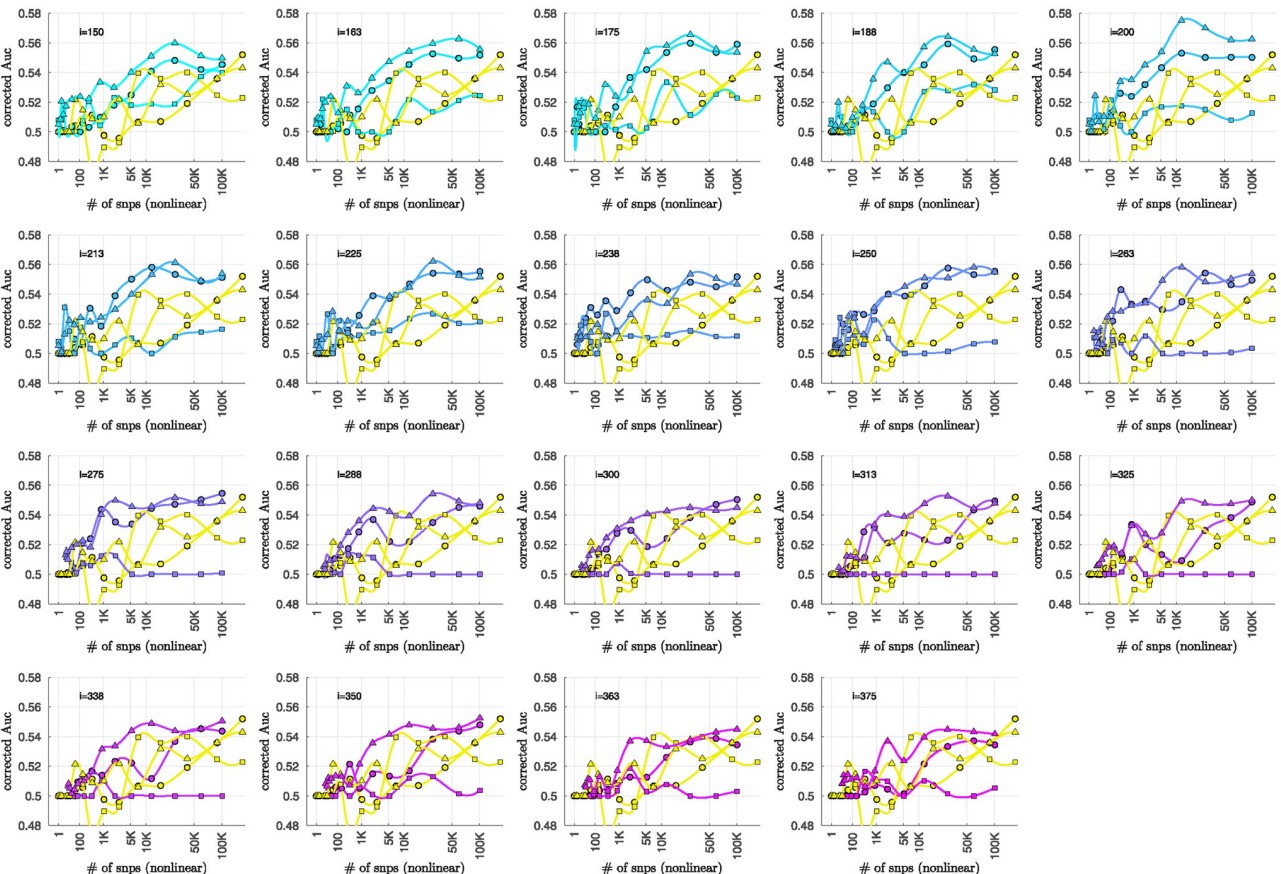

**Fig 10. This figure is similar to Fig 8, except that we limit ourselves only to those case-subjects in the replication-arms which are classified as BDII.** This subset corresponded to 19% ($M = 1082$), 12% ($M = 435$) and 16% ($M = 1060$) of the case-population for arms 2, 3 and 4, respectively. The corresponding **AUC**-values are denoted by $\mathbf{AUC}_{\mathrm{wide|BDII}}(\tilde{p})$ and $\mathbf{AUC}_{\mathrm{bicl|BDII}}(\tilde{p})$ in the main text. For reference the training-arm had $M = 788$ BDII case-subjects, corresponding to 31% of the case-population in arm-1.

We believe that Fig 11 hints at the potential our methodology offers to researchers of complex disease. By limiting our definition of a case to those with a more genetically homogeneous BD signature, we were able to generate a $\mathbf{PRS}_{\mathrm{bicl}}$ which outperforms $\mathbf{PRS}_{\mathrm{wide}}$ in the following ways:

- The maximum $\mathcal{R}^2_{\mathrm{bicl}}$ is 20–40% higher than the maximum $\mathcal{R}^2_{\mathrm{wide}}$, depending on the iteration-index $i$.

- This increase in liability-score occurs despite the fact that the $\mathbf{PRS}_{\mathrm{bicl}}$ is generated using $\sim$ 50% to 80% fewer cases than the $\mathbf{PRS}_{\mathrm{wide}}$. For example, we considered only between 1191–526 cases in arm-1 to generate the $\mathbf{PRS}_{\mathrm{bicl}}(j'; i, \tilde{p})$ values for $i = [175, 350]$, whereas 2524 cases were used to generate the $\mathbf{PRS}_{\mathrm{wide}}(j'; \tilde{p})$ values.

- The p-values assigned to the SNPs via the bicluster-informed GWAS were less noisy than those p-values assigned using the population-wide GWAS. For example, Fig 11 indicates that the first 10K SNPs of highest significance from the population-wide GWAS contained almost no disease-related information. By contrast, the first 10K SNPs of highest significance from the bicluster-informed GWAS typically contain most of the available disease-related

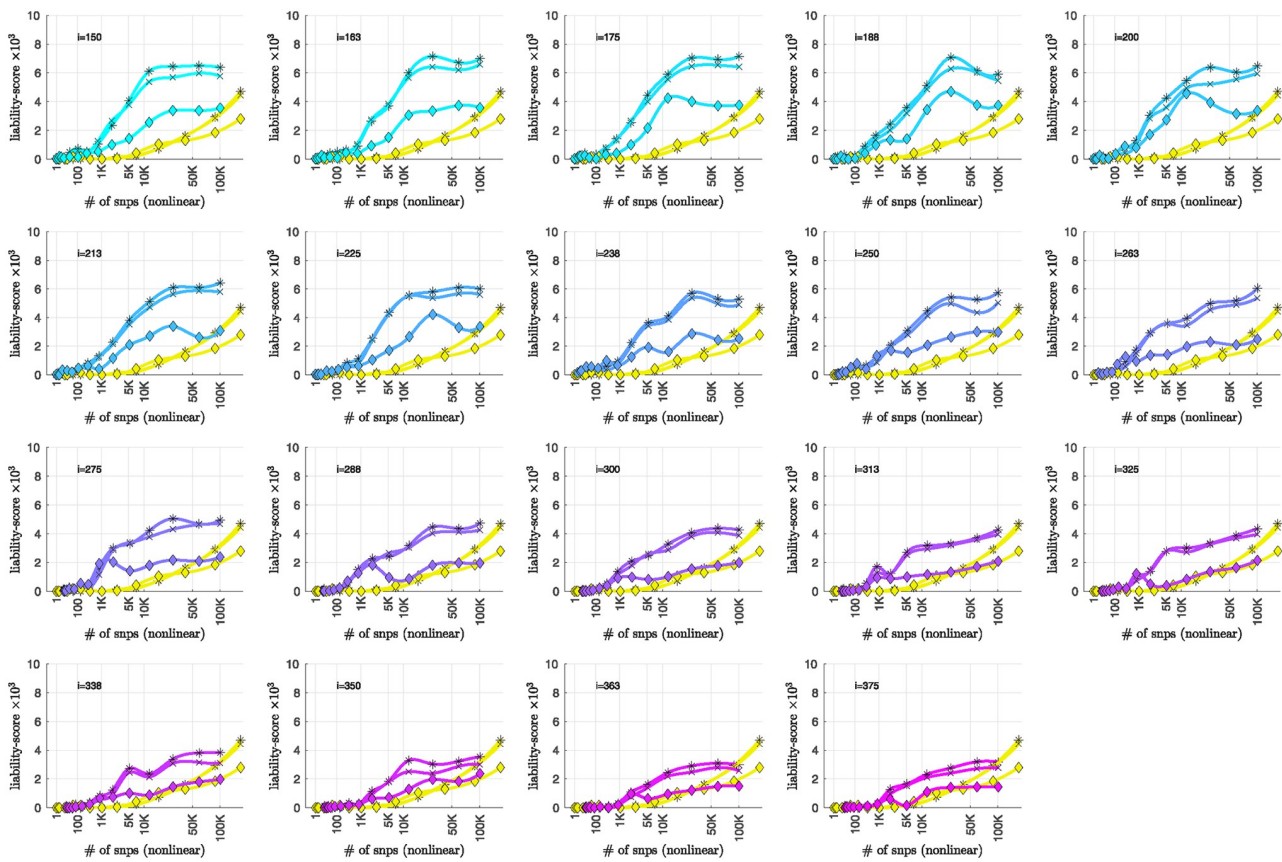

**Fig 11. This figure is similar to Fig 8, and uses the data from Figs 8, 9 and 10.** This time we combine the information across all three replication-arms, and calculate replication **AUC**-values for this combined data-set. We then convert these **AUC**-values into liability-scores (see [71]). The results for all the cases ($\mathcal{R}^2_{\text{wide}}$ and $\mathcal{R}^2_{\text{bicl}}$) are shown with an asterisk '*', whereas the results for only the BD1-cases ($\mathcal{R}^2_{\text{wide|BDI}}$ and $\mathcal{R}^2_{\text{bicl|BDI}}$) are shown with an '×', and the results for only the BD2-cases ($\mathcal{R}^2_{\text{wide|BDII}}$ and $\mathcal{R}^2_{\text{bicl|BDII}}$) are shown with a diamond. In each case the yellow curves correspond to the liability-scores derived from the population-wide PRS, whereas the cyan-magenta curves correspond to the liability-scores derived from the bicluster-informed PRS. Note that our overall results are closely matched by the BD1-cases, but not by the BD2-cases.

information. The bicluster-informed GWAS produces a $\mathcal{R}^2_{\text{bicl}}$ with only $\sim$ 5K to 10K SNPs that surpasses the maximum of $\mathcal{R}^2_{\text{wide}}$ (e.g., within the $i = 225$ subplot the value of $\mathcal{R}^2_{\text{bicl}}$ at 5K SNPs is comparable to the value of $\mathcal{R}^2_{\text{wide}}$ at $\sim$ 150K SNPs).

- Furthermore, the values of $\mathcal{R}^2_{\text{bicl}}(i, \tilde{p})$ typically plateau somewhere between 10K–35K SNPs (corresponding to $\tilde{p} \in [0.22, 0.36]$). Meanwhile, the values of $\mathcal{R}^2_{\text{wide}}(\tilde{p})$ continue to increase until $\tilde{p} = 1$ (including all $\sim$ 150K SNPs).

To summarize: The **PRS**$_{\text{bicl}}$ outperforms **PRS**$_{\text{wide}}$ overall, and when restricted to either BDI and BDII, achieving a higher maximum for each subtype. The **PRS**$_{\text{bicl}}$ also achieves its peak performance with far fewer SNPs, consistent with a far less noisy signal. Put another way, the values of $\mathcal{R}^2_{\text{bicl}}$, $\mathcal{R}^2_{\text{bicl|BDI}}$ and $\mathcal{R}^2_{\text{bicl|BDII}}$ all plateau earlier than $\mathcal{R}^2_{\text{wide}}$, $\mathcal{R}^2_{\text{wide|BDI}}$ and $\mathcal{R}^2_{\text{wide|BDII}}$, indicating that the SNPs which are most relevant to the bicluster-informed PRS performance have indeed been identified by the bicluster-informed GWAS as having low individual $p$-values.

Additionally, we note that there is a close relationship between $\mathcal{R}^2_{\text{bicl}}$ and $\mathcal{R}^2_{\text{bicl|BDI}}$, but a discrepancy between $\mathcal{R}^2_{\text{bicl}}$ and $\mathcal{R}^2_{\text{bicl|BDII}}$. The values of $\mathcal{R}^2_{\text{bicl|BDII}}$ indicate that some subset of BDII cases share the bicluster signature, but the maximum for $\mathcal{R}^2_{\text{bicl|BDII}}$ is only 50% of the maximum for $\mathcal{R}^2_{\text{bicl|BDI}}$. This could imply that the bicluster has focused on a signature that is associated with BDI, perhaps serving as a risk factor for manic episodes in the presence of the necessary epigenetic or environmental influences.

## Gene-enrichment

We also perform a simple over-representation analysis, measuring the overlap $\kappa(i, l)$ between the bicluster $D(i)$ at iteration $i$ and the various neuronally-related pathways $\mathcal{H}_l$ from the `go_bp` ontology (see Methods). The average z-score for the enrichment-values $\kappa(i, l)$, averaged over the interval $i \in [175, 350]$ and all neuronally-related pathways, is quite significant, with $p \lesssim 1e - 4$ (as determined by a permutation-test). Examples of some of the more significantly over-represented pathways are shown in Table 1.

## Secondary bicluster

After discovering and analyzing the primary bicluster within arm-1 (described above), we searched for a secondary bicluster. We first eliminated the structure associated with the primary bicluster by scrambling the entries of the submatrix $D(175)$ (see [64] for details). We then reran our half-loop algorithm on this scrambled version of arm-1. While we did find a secondary trace that was indicative of heterogeneity, the overall level of differential-expression was far lower than for the first bicluster (see Fig 25 in S1 Text). Moreover, the structure associated with this secondary trace did not significantly replicate (see Figs 26–28 in S1 Text). It is possible that a secondary bicluster exists, but that we could not pinpoint it due to a lack of power in our training-arm. It is also possible that the scrambled version of arm-1 is heterogeneous, but not in a way that can be described by a bicluster (see [64] for examples along these lines). In either case, a larger sample size will be required to further probe this residual heterogeneity.

## Control biclusters

Up to this point we have only considered biclusters within the case-population; i.e., subsets of case-subjects which exhibit a genetic-signature that is not shared by the control-subjects. It is natural to ask if there are also biclusters that exist within the control-population (i.e., whether or not the control-population is homogeneous). Such 'control-biclusters' might be induced by batch effects or issues associated with recruitment; e.g., many of the BD controls may be drawn from another disease study (such as cancer), thus being more likely to share certain genetic features. It might also be the case that some of the control-biclusters are biologically significant, corresponding to mechanisms which protect against the disease. In either scenario, a better understanding of the heterogeneity within the control-population can assist in designing homogeneous populations of controls for future studies.

We can easily carry out this analysis simply by reversing the labels within our biclustering algorithm (i.e., swapping $D$ and $X$). This reversed search will find biclusters that are driven by genetic-signatures which are more prevalent within the controls than within the cases. As mentioned above, we find that the control-population within arm-1 is quite homogeneous: the trace decays monotonically with no distinguished peaks (see Fig 29 in S1 Text). This homogeneity can be viewed as a validation of our initial choice of arm-1 as a training- or discovery-arm.

**Table 1. Here we list some of the pathways from the go_bp ontology.** Shown here are only the 32 most significant pathways as determined by $\kappa(175, l)$. Each pathway is listed alongside approximations to its individual over-representation p-value (estimated using the hypergeometric-distribution). The $-\log_{10}(p)$-values are listed for iterations 175–350 (see top row). Those annotations with an individual over-representation p-value smaller than 0.05 are in bold.

| annotation | 175 | 200 | 225 | 250 | 275 | 300 | 325 | 350 | 375 |
|---|---|---|---|---|---|---|---|---|---|
| synaptic vesicle endocytosis | **3.49** | **2.10** | 1.28 | **1.94** | 0.22 | 0.24 | 0.24 | 0.26 | 0.26 |
| positive regulation of neurogenesis | **3.29** | **2.36** | 0.36 | 0.39 | 0.75 | 0.08 | 0.11 | 0.13 | 0.15 |
| neurological system process involved in regulation | **3.10** | **2.88** | **2.10** | 1.07 | **1.46** | **1.74** | 0.21 | 0.22 | 0.23 |
| positive regulation of neuroblast proliferation | **2.97** | 0.71 | 0.18 | 0.20 | 0.22 | 0.23 | 0.24 | 0.26 | 0.30 |
| neurological system process | **2.94** | 1.10 | **1.78** | **2.14** | 1.11 | 0.84 | 0.43 | 0.98 | 1.20 |
| synaptic vesicle exocytosis | **2.86** | **3.03** | **3.62** | **4.31** | **2.22** | 0.17 | 0.19 | 0.19 | 0.19 |
| regulation of neurogenesis | **2.81** | **1.57** | 0.85 | 0.60 | 0.96 | 0.36 | 0.68 | 0.53 | 0.89 |
| establishment of synaptic vesicle localization | **2.78** | **2.03** | **2.70** | **3.66** | **1.85** | 0.15 | 0.17 | 0.18 | 0.18 |
| synaptic vesicle localization | **2.78** | **2.03** | **2.70** | **3.66** | **1.85** | 0.15 | 0.17 | 0.18 | 0.18 |
| synaptic vesicle transport | **2.78** | **2.03** | **2.70** | **3.66** | **1.85** | 0.15 | 0.17 | 0.18 | 0.18 |
| positive regulation of neuron differentiation | **2.63** | 0.99 | 0.47 | 0.34 | 0.54 | 0.87 | 1.27 | 0.16 | 0.20 |
| axonogenesis | **2.53** | **1.83** | **2.41** | 1.03 | **1.73** | 1.30 | 1.03 | 1.03 | 0.85 |
| cell morphogenesis involved in neuron differentiat | **2.52** | **2.18** | **3.19** | **1.44** | **2.42** | **1.67** | **1.45** | **1.53** | **1.51** |
| generation of neurons | **2.31** | **1.98** | **2.54** | **1.46** | **2.06** | 1.07 | 0.98 | 1.06 | **1.36** |
| axon development | **2.25** | **1.70** | **2.29** | 0.99 | **1.68** | 1.26 | 0.99 | 1.00 | 0.83 |
| positive regulation of axonogenesis | **2.17** | **3.36** | 0.64 | 0.43 | 0.64 | 0.13 | 0.15 | 0.18 | 0.19 |
| axonal fasciculation | **2.07** | 0.17 | 0.21 | 0.24 | 0.30 | 0.30 | 0.30 | 0.30 | 0.30 |
| neuron development | **2.04** | **1.73** | **2.73** | **2.08** | **2.62** | **1.32** | 0.95 | 1.25 | 1.30 |
| central nervous system projection neuron axonogene | **2.04** | **1.76** | **2.94** | 0.16 | 0.20 | 0.21 | 0.23 | 0.25 | 0.25 |
| neurogenesis | **2.01** | **1.64** | **2.38** | **1.57** | **2.42** | 0.90 | 0.86 | 0.92 | 1.16 |
| neuron projection morphogenesis | **1.91** | **1.53** | **2.23** | 1.13 | **1.96** | 1.18 | 0.92 | 0.90 | 0.72 |
| central nervous system neuron axonogenesis | **1.88** | **1.85** | **3.40** | 0.12 | 0.16 | 0.17 | 0.19 | 0.22 | 0.23 |
| neurotransmitter uptake | **1.87** | 0.41 | 0.12 | 0.15 | 0.17 | 0.18 | 0.20 | 0.22 | 0.24 |
| neuron projection development | **1.81** | 1.17 | **2.33** | **1.57** | **1.84** | 1.02 | 0.90 | 1.02 | 0.93 |
| axon guidance | **1.81** | 0.83 | **1.64** | 1.11 | **1.63** | **2.05** | **2.18** | **1.80** | **1.58** |
| neuron projection guidance | **1.81** | 0.83 | **1.64** | 1.11 | **1.63** | **2.05** | **2.18** | **1.80** | **1.58** |
| synapsis | **1.69** | 1.04 | **1.97** | **3.19** | 0.18 | 0.20 | 0.23 | 0.24 | 0.25 |
| synaptic transmission | **1.64** | 0.51 | 0.40 | 0.77 | 0.44 | 0.26 | 0.31 | 0.59 | 0.44 |
| regulation of neurological system process | **1.63** | 0.90 | 1.06 | **1.37** | 0.53 | 0.45 | 0.78 | 1.06 | 0.56 |
| neuron differentiation | **1.59** | **1.40** | **2.04** | **1.73** | **2.20** | **1.34** | 1.21 | 1.25 | **1.53** |
| positive regulation of neurological system process | **1.48** | 0.61 | 0.67 | 0.46 | 0.11 | 0.13 | 0.17 | 0.19 | 0.20 |

On the other hand, we find strong evidence for heterogeneity within the control-populations of arms 2, 3 and 4 (see Figs 30—32 in S1 Text). In each case the trace has a significant distinguished maximum involving only a fraction of the control-subjects (i.e,. 13%, 28% and 15% of the controls, respectively).

The heterogeneity observed in the control-populations of arms 2, 3 and 4 might be expected; each of these arms comprises multiple smaller studies. Notably however, the 'control-biclusters' within these arms cannot all be easily dismissed as batch-effects. Indeed, each of the dominant control-biclusters is also quite significant, while also usually well balanced across the ancestry-related covariates and individual cohorts within each arm. Each of these dominant control-biclusters also replicates across the majority of other arms.

Thus, while a portion of these control-biclusters might be driven by batch-effects or other idiosyncrasies in the control-population, it is possible that some of these signals have biological relevance, perhaps involving mechanisms which protect against BD (as the control-biclusters

were identified specifically because they involved genetic patterns not as prevalent across the cases). Consequently, we would recommend considering this heterogeneity when performing other kinds of analysis. For example, one should not necessarily assume that the controls are homogeneous, as small subgroups of controls can likely exhibit genetic-signatures that are distinct from the rest.

## Conclusion

In this paper we have taken a 'genotype-driven' approach to investigating genotypic-heterogeneity. That is to say, first we used only basic phenotypic classification to divide subjects into cases (BD) and controls (not BD). We then applied a biclustering analysis to identify genetic subgroups within the case-population. Analyzing the BDI and BDII cases together as a whole allowed us to identify a genetic subgroup (i.e., the bicluster described above). This bicluster involved a genetically homogeneous subset of the BD-cases within the training-arm, which we then used to inform a more robust PRS with better replication across studies.

Our results suggest two hypotheses for future work. Most directly, our replication- and PRS-analyses indicate that the bicluster we found within the training-arm indeed represents a genetic subgroup of BD which generalizes across data-sets. More generally, our results provide a proof-of-principle for our overall methodology: a data-driven approach to identifying genetically homogeneous subsets of case-subjects can help construct more robust PRSs, with the potential of improving SNP-replication in BD GWAS and, ultimately, a better understanding of the etiology of Bipolar Disorder.

In some respects our approach can be termed 'unsupervised', as we did not use BD-subtype (BDI vs. BDII) or subphenotype information to guide our primary analysis. This unsupervised approach allows us to circumvent many of the challenges associated with phenotype classification, such as missingness and variation in assessment and collection process (e.g., expert-led vs. self-report). It also allows us to identify genetic patterns which straddle traditional classifications provided the signature is not present in the control group. E.g., though our bicluster was enriched for BDI, it was by no means limited to BDI and included many BDII cases.

Along these lines, we believe that a similar unsupervised approach could be used to search for interactions between the signals we have found and other diseases, as well as for cross-psychiatric-disorder signals not present in the control group. There are many examples of genetic interactions along these lines: the SNPs driving BD have a strong correlation with those driving schizophrenia, and also share overlap with the SNPs driving MDD, OCD, anorexia nervosa, ADHD, ASD and substance-abuse [34, 73, 74]. Many SNPs have also been associated with other disorders [17, 75–77]. More generally speaking, BD shows substantial overlap with other disorders; e.g., more than 90% of BD subjects exhibit lifetime comorbidity [3] with at least one other psychiatric disorder [58, 78, 79], or non-psychiatric disorder [80–82]. This high rate of comorbidity implies that BD is one of multiple disorders which perturb several important regulatory systems [83, 84]. Given these relationships, it is possible that the bicluster-score and/or the bicluster-limited PRSs may also correlate with some of the signals of these other disorders. It is possible that we could discover interesting biclusters which cross psychiatric disorders or are present in the control groups and predict resistance to psychiatric illness more generally; we defer an investigation of these interactions to future work.

The biclustering algorithm we use also offers a 'supervised' option which uses additional information (e.g., BD-subtype or other clinical data) to subdivide the case-population while searching for heterogeneity. Sex might be one important variable to include in such a supervised BD analysis. For example, while most studies do not indicate large difference in BD prevalence between men and women (indeed, the bicluster we identified was not significantly

enriched for sex), there is some evidence of a sex disparity in the prevalence of BDII, rapid-cycling and mixed-episodes [85, 86]. Age may also be an important role-player, as an earlier age of onset may be associated with higher severity and a poorer long-term prognosis (possibly due to mis-diagnoses at an early stage) [57, 87].

One limitation of our current study is that it is restricted to common variants (i.e., SNPs with a high minor-allele-frequency). While it is encouraging that the common variants alone can be used to find replicable and robust signals, it is also likely that the rare variants also play a role in the heterogeneity of BD. Analyzing the rare variants brings new challenges, as rare variants often require more statistical power to detected and/or validate [88–93].

Another more serious limitation is that our training-arm is quite restricted in terms of ancestry. More generally, almost all the individuals in our data-set are of European descent. We expect that this lack of diversity will limit our ability to pinpoint the most biologically relevant signals, as many previous GWAS analyses have not generalized well to cohorts of different ancestry [29, 94–98]. An important future direction will be to investigate the interactions between genotypic heterogeneity and ancestry.

We do not expect a full analysis of genetic-heterogeneity to be entirely trivial. For example, appropriately correcting for ancestry is not always easy, even when searching for homogeneous signals. When searching for heterogeneity such a correction becomes more complicated and, necessarily, involves more parameters. Larger (and more diverse) sample sizes will likely be necessary to clarify (i) the disease-specific genetic-subgroups (i.e., biclusters) within BD, as well as (ii) the phenotypic subtypes of BD, and perhaps most importantly: (iii) the interaction between these subgroups and subtypes and other covariates such as ancestry. We suspect that a careful treatment of the associated statistical issues will pose a significant challenge. Nevertheless, these advancements will likely further improve our understanding of the etiology of BD.

## Supporting information

**S1 Text. Contains a detailed description of our methods, including an outline of the steps involved and the considerations we made along the way.** Also contains the supporting figures referenced in the main text.
(PDF)

## Acknowledgments

The chair of the Bipolar Disorder Working Group of the Psychiatric Genomics Consortium is currently Ole A Andreassen (contact email: o.a.andreassen@medisin.uio.no). The contributors to the Bipolar Disorder Working Group are listed below.

### Bipolar Disorder Working Group of the Psychiatric Genomics Consortium

Eli A Stahl[1,2,3], Gerome Breen[4,5], Andreas J Forstner[6,7,8,9,10], Andrew McQuillin[11], Stephan Ripke[12,13,14], Vassily Trubetskoy[13], Manuel Mattheisen[15,16,17,18,19], Yunpeng Wang[20,21], Jonathan R I Coleman[4,5], Héléna A Gaspar[4,5], Christiaan A de Leeuw[22], Stacy Steinberg[23], Jennifer M Whitehead Pavlides[24], Maciej Trzaskowski[25], Enda M Byrne[25], Tune H Pers[3,26], Peter A Holmans[27], Alexander L Richards[27], Liam Abbott[12], Esben Agerbo[19,28,29], Huda Akil[30], Diego Albani[31], Ney Alliey-Rodriguez[32], Thomas D Als[15,16,19], Adebayo Anjorin[33], Verneri Antilla[14], Swapnil Awasthi[13], Judith A Badner[34], Marie Bækvad-Hansen[19,35], Jack D Barchas[36], Nicholas Bass[11], Michael Bauer[37], Richard Belliveau[12], Sarah E Bergen[38], Carsten Bøcker

Pedersen[19,28,29], Erlend Bøen[39], Marco P. Boks[40], James Boocock[41], Monika Budde[42], William Bunney[43], Margit Burmeister[44], Jonas Bybjerg-Grauholm[19,35], William Byerley[45], Miquel Casas[46,47,48,49], Felecia Cerrato[12], Pablo Cervantes[50], Kimberly Chambert[12], Alexander W Charney[2], Danfeng Chen[12], Claire Churchhouse[12,14], Toni-Kim Clarke[51], William Coryell[52], David W Craig[53], Cristiana Cruceanu[50,54], David Curtis[55,56], Piotr M Czerski[57], Anders M Dale[58,59,60,61], Simone de Jong[4,5], Franziska Degenhardt[8], Jurgen Del-Favero[62], J Raymond DePaulo[63], Srdjan Djurovic[64,65], Amanda L Dobbyn[1,2], Ashley Dumont[12], Torbjørn Elvsåshagen[66,67], Valentina Escott-Price[27], Chun Chieh Fan[61], Sascha B Fischer[6,10], Matthew Flickinger[68], Tatiana M Foroud[69], Liz Forty[27], Josef Frank[70], Christine Fraser[27], Nelson B Freimer[71], Louise Frisén[72,73,74], Katrin Gade[42,75], Diane Gage[12], Julie Garnham[76], Claudia Giambartolomei[206], Marianne Giørtz Pedersen[19,28,29], Jaqueline Goldstein[12], Scott D Gordon[77], Katherine Gordon-Smith[78], Elaine K Green[79], Melissa J Green[80,133], Tiffany A Greenwood[60], Jakob Grove[15,16,19,81], Weihua Guan[82], José Guzman-Parra[83], Marian L Hamshere[27], Martin Hautzinger[84], Urs Heilbronner[42], Stefan Herms[6,8,10], Maria Hipolito[85], Per Hoffmann[6,8,10], Dominic Holland[58,86], Laura Huckins[1,2], Stéphane Jamain[87,88], Jessica S Johnson[1,2], Radhika Kandaswamy[4], Robert Karlsson[38], James L Kennedy[89,90,91,92], Sarah Kittel-Schneider[93], James A Knowles[94,95], Manolis Kogevinas[96], Anna C Koller[8], Ralph Kupka[97,98,99], Catharina Lavebratt[72], Jacob Lawrence[100], William B Lawson[85], Markus Leber[101], Phil H Lee[12,14,102], Shawn E Levy[103], Jun Z Li[104], Chunyu Liu[105], Susanne Lucae[106], Anna Maaser[8], Donald J MacIntyre[107,108], Pamela B Mahon[63,109], Wolfgang Maier[110], Lina Martinsson[73], Steve McCarroll[12,111], Peter McGuffin[4], Melvin G McInnis[112], James D McKay[113], Helena Medeiros[95], Sarah E Medland[77], Fan Meng[30,112], Lili Milani[114], Grant W Montgomery[25], Derek W Morris[115,116], Thomas W Mühleisen[6,117], Niamh Mullins[4], Hoang Nguyen[1,2], Caroline M Nievergelt[60,118], Annelie Nordin Adolfsson[119], Evaristus A Nwulia[85], Claire O'Donovan[76], Loes M Olde Loohuis[71], Anil P S Ori[71], Lilijana Oruc[120], Urban Ösby[121], Roy H Perlis[122,123], Amy Perry[78], Andrea Pfennig[37], James B Potash[63], Shaun M Purcell[2,109], Eline J Regeer[124], Andreas Reif[93], Céline S Reinbold[6,10], John P Rice[125], Fabio Rivas[83], Margarita Rivera[4,126], Panos Roussos[1,2,127], Douglas M Ruderfer[128], Euijung Ryu[129], Cristina Sánchez-Mora[46,47,49], Alan F Schatzberg[130], William A Scheftner[131], Nicholas J Schork[132], Cynthia Shannon Weickert[80,133], Tatyana Shehktman[60], Paul D Shilling[60], Engilbert Sigurdsson[134], Claire Slaney[76], Olav B Smeland[135,136], Janet L Sobell[137], Christine Søholm Hansen[19,35], Anne T Spijker[138], David St Clair[139], Michael Steffens[140], John S Strauss[91,141], Fabian Streit[70], Jana Strohmaier[70], Szabolcs Szelinger[142], Robert C Thompson[112], Thorgeir E Thorgeirsson[23], Jens Treutlein[70], Helmut Vedder[143], Weiqing Wang[1,2], Stanley J Watson[112], Thomas W Weickert[80,133], Stephanie H Witt[70], Simon Xi[144], Wei Xu[145,146], Allan H Young[147], Peter Zandi[148], Peng Zhang[149], Sebastian Zöllner[112], eQTLGen Consortium, BIOS Consortium, Rolf Adolfsson[119], Ingrid Agartz[17,39,150], Martin Alda[76,151], Lena Backlund[73], Bernhard T Baune[152,158], Frank Bellivier[153,154,155,156], Wade H Berrettini[157], Joanna M Biernacka[129], Douglas H R Blackwood[51], Michael Boehnke[68], Anders D Børglum[15,16,19], Aiden Corvin[116], Nicholas Craddock[27], Mark J Daly[12,14], Udo Dannlowski[158], Tõnu Esko[3,111,114,159], Bruno Etain[153,155,156,160], Mark Frye[161], Janice M Fullerton[133,162], Elliot S Gershon[32,163], Michael Gill[116], Fernando Goes[63], Maria Grigoroiu-Serbanescu[164], Joanna Hauser[57], David M Hougaard[19,35], Christina M Hultman[38], Ian Jones[27], Lisa A Jones[78], René S Kahn[2,40], George Kirov[27], Mikael Landén[38,165], Marion Leboyer[88,153,166], Cathryn M Lewis[4,5,167], Qingqin S Li[168], Jolanta Lissowska[169], Nicholas G Martin[77,170], Fermin Mayoral[83], Susan L McElroy[171], Andrew M McIntosh[51,172], Francis J McMahon[173], Ingrid Melle[174,175], Andres Metspalu[114,176], Philip B Mitchell[80], Gunnar Morken[177,178], Ole Mors[19,179], Preben Bo Mortensen[15,19,28,29], Bertram Müller-Myhsok[54,180,181], Richard M Myers[103], Benjamin M Neale[3,12,14], Vishwajit Nimgaonkar[182], Merete Nordentoft[19,183], Markus M Nöthen[8], Michael C O'Donovan[27], Ketil J Oedegaard[184,185], Michael J

Owen[27], Sara A Paciga[186], Carlos Pato[95,187], Michele T Pato[95], Danielle Posthuma[22,188], Josep Antoni Ramos-Quiroga[46,47,48,49], Marta Ribasés[46,47,49], Marcella Rietschel[70], Guy A Rouleau[189,190], Martin Schalling[72], Peter R Schofield[133,162], Thomas G Schulze[42,63,70,75,173], Alessandro Serretti[191], Jordan W Smoller[12,192,193], Hreinn Stefansson[23], Kari Stefansson[23,194], Eystein Stordal[195,196], Patrick F Sullivan[38,197,198], Gustavo Turecki[199], Arne E Vaaler[200], Eduard Vieta[201], John B Vincent[141], Thomas Werge[19,202,203], John I Nurnberger[204], Naomi R Wray[24,25], Arianna Di Florio[27,198], Howard J Edenberg[205], Sven Cichon[6,8,10,117], Roel A Ophoff[40,41,71], Laura J Scott[68], Ole A Andreassen[135,136], John Kelsoe[60], Pamela Sklar[1,2,†]

[1]Department of Genetics and Genomic Sciences, Icahn School of Medicine at Mount Sinai, New York, NY, US. [2]Department of Psychiatry, Icahn School of Medicine at Mount Sinai, New York, NY, US. [3]Medical and Population Genetics, Broad Institute, Cambridge, MA, US. [4]MRC Social, Genetic and Developmental Psychiatry Centre, King's College London, London, GB. [5]NIHR BRC for Mental Health, King's College London, London, GB. [6]Department of Biomedicine, University of Basel, Basel, CH. [7]Department of Psychiatry (UPK), University of Basel, Basel, CH. [8]Institute of Human Genetics, University of Bonn, School of Medicine & University Hospital Bonn, Bonn, DE. [9]Centre for Human Genetics, University of Marburg, Marburg, DE. [10]Institute of Medical Genetics and Pathology, University Hospital Basel, Basel, CH. [11]Division of Psychiatry, University College London, London, GB. [12]Stanley Center for Psychiatric Research, Broad Institute, Cambridge, MA, US. [13]Department of Psychiatry and Psychotherapy, Charité—Universitätsmedizin, Berlin, DE. [14]Analytic and Translational Genetics Unit, Massachusetts General Hospital, Boston, MA, US. [15]iSEQ, Center for Integrative Sequencing, Aarhus University, Aarhus, DK. [16]Department of Biomedicine—Human Genetics, Aarhus University, Aarhus, DK. [17]Department of Clinical Neuroscience, Centre for Psychiatry Research, Karolinska Institutet, Stockholm, SE. [18]Department of Psychiatry, Psychosomatics and Psychotherapy, Center of Mental Health, University Hospital Würzburg, Würzburg, DE. [19]iPSYCH, The Lundbeck Foundation Initiative for Integrative Psychiatric Research, DK. [20]Institute of Biological Psychiatry, Mental Health Centre Sct. Hans, Copenhagen, DK. [21]Institute of Clinical Medicine, University of Oslo, Oslo, NO. [22]Department of Complex Trait Genetics, Center for Neurogenomics and Cognitive Research, Amsterdam Neuroscience, Vrije Universiteit Amsterdam, Amsterdam, NL. [23]deCODE Genetics / Amgen, Reykjavik, IS. [24]Queensland Brain Institute, The University of Queensland, Brisbane, QLD, AU. [25]Institute for Molecular Bioscience, The University of Queensland, Brisbane, QLD, AU. [26]Division of Endocrinology and Center for Basic and Translational Obesity Research, Boston Children's Hospital, Boston, MA, US. [27]Medical Research Council Centre for Neuropsychiatric Genetics and Genomics, Division of Psychological Medicine and Clinical Neurosciences, Cardiff University, Cardiff, GB. [28]National Centre for Register-Based Research, Aarhus University, Aarhus, DK. [29]Centre for Integrated Register-based Research, Aarhus University, Aarhus, DK. [30]Molecular & Behavioral Neuroscience Institute, University of Michigan, Ann Arbor, MI, US. [31]Department of Neuroscience, IRCCS—Istituto Di Ricerche Farmacologiche Mario Negri, Milan, IT. [32]Department of Psychiatry and Behavioral Neuroscience, University of Chicago, Chicago, IL, US. [33]Psychiatry, Berkshire Healthcare NHS Foundation Trust, Bracknell, GB. [34]Psychiatry, Rush University Medical Center, Chicago, IL, US. [35]Center for Neonatal Screening, Department for Congenital Disorders, Statens Serum Institut, Copenhagen, DK. [36]Department of Psychiatry, Weill Cornell Medical College, New York, NY, US. [37]Department of Psychiatry and Psychotherapy, University Hospital Carl Gustav Carus, Technische Universität Dresden, Dresden, DE. [38]Department of Medical Epidemiology and Biostatistics, Karolinska Institutet, Stockholm, SE. [39]Department of Psychiatric Research, Diakonhjemmet Hospital, Oslo, NO. [40]Psychiatry, UMC Utrecht Brain Center Rudolf Magnus, Utrecht, NL. [41]Human Genetics, University of California Los Angeles, Los Angeles, CA,

US. [42]Institute of Psychiatric Phenomics and Genomics (IPPG), University Hospital, LMU Munich, Munich, DE. [43]Department of Psychiatry and Human Behavior, University of California, Irvine, Irvine, CA, US. [44]Molecular & Behavioral Neuroscience Institute and Department of Computational Medicine & Bioinformatics, University of Michigan, Ann Arbor, MI, US. [45]Psychiatry, University of California San Francisco, San Francisco, CA, US. [46]Instituto de Salud Carlos III, Biomedical Network Research Centre on Mental Health (CIBERSAM), Madrid, ES. [47]Department of Psychiatry, Hospital Universitari Vall d´Hebron, Barcelona, ES. [48]Department of Psychiatry and Forensic Medicine, Universitat Autònoma de Barcelona, Barcelona, ES. [49]Psychiatric Genetics Unit, Group of Psychiatry Mental Health and Addictions, Vall d´Hebron Research Institut (VHIR), Universitat Autònoma de Barcelona, Barcelona, ES. [50]Department of Psychiatry, Mood Disorders Program, McGill University Health Center, Montreal, QC, CA. [51]Division of Psychiatry, University of Edinburgh, Edinburgh, GB. [52]University of Iowa Hospitals and Clinics, Iowa City, IA, US. [53]Translational Genomics, USC, Phoenix, AZ, US. [54]Department of Translational Research in Psychiatry, Max Planck Institute of Psychiatry, Munich, DE. [55]Centre for Psychiatry, Queen Mary University of London, London, GB. [56]UCL Genetics Institute, University College London, London, GB. [57]Department of Psychiatry, Laboratory of Psychiatric Genetics, Poznan University of Medical Sciences, Poznan, PL. [58]Department of Neurosciences, University of California San Diego, La Jolla, CA, US. [59]Department of Radiology, University of California San Diego, La Jolla, CA, US. [60]Department of Psychiatry, University of California San Diego, La Jolla, CA, US. [61]Department of Cognitive Science, University of California San Diego, La Jolla, CA, US. [62]Applied Molecular Genomics Unit, VIB Department of Molecular Genetics, University of Antwerp, Antwerp, Belgium. [63]Department of Psychiatry and Behavioral Sciences, Johns Hopkins University School of Medicine, Baltimore, MD, US. [64]Department of Medical Genetics, Oslo University Hospital Ullevål, Oslo, NO. [65]NORMENT, KG Jebsen Centre for Psychosis Research, Department of Clinical Science, University of Bergen, Bergen, NO. [66]Department of Neurology, Oslo University Hospital, Oslo, NO. [67]NORMENT, KG Jebsen Centre for Psychosis Research, Oslo University Hospital, Oslo, NO. [68]Center for Statistical Genetics and Department of Biostatistics, University of Michigan, Ann Arbor, MI, US. [69]Department of Medical & Molecular Genetics, Indiana University, Indianapolis, IN, US. [70]Department of Genetic Epidemiology in Psychiatry, Central Institute of Mental Health, Medical Faculty Mannheim, Heidelberg University, Mannheim, DE. [71]Center for Neurobehavioral Genetics, University of California Los Angeles, Los Angeles, CA, US. [72]Department of Molecular Medicine and Surgery, Karolinska Institutet and Center for Molecular Medicine, Karolinska University Hospital, Stockholm, SE. [73]Department of Clinical Neuroscience, Karolinska Institutet and Center for Molecular Medicine, Karolinska University Hospital, Stockholm, SE. [74]Child and Adolescent Psychiatry Research Center, Stockholm, SE. [75]Department of Psychiatry and Psychotherapy, University Medical Center Göttingen, Göttingen, DE. [76]Department of Psychiatry, Dalhousie University, Halifax, NS, CA. [77]Genetics and Computational Biology, QIMR Berghofer Medical Research Institute, Brisbane, QLD, AU. [78]Department of Psychological Medicine, University of Worcester, Worcester, GB. [79]School of Biomedical Sciences, Plymouth University Peninsula Schools of Medicine and Dentistry, University of Plymouth, Plymouth, GB. [80]School of Psychiatry, University of New South Wales, Sydney, NSW, AU. [81]Bioinformatics Research Centre, Aarhus University, Aarhus, DK. [82]Biostatistics, University of Minnesota System, Minneapolis, MN, US. [83]Mental Health Department, University Regional Hospital, Biomedicine Institute (IBIMA), Málaga, ES. [84]Department of Psychology, Eberhard Karls Universität Tübingen, Tubingen, DE. [85]Department of Psychiatry and Behavioral Sciences, Howard University Hospital, Washington, DC, US. [86]Center for Multimodal Imaging and Genetics, University of California San Diego, La Jolla, CA, US. [87]Psychiatrie Translationnelle, Inserm U955, Créteil, FR.

[88]Faculté de Médecine, Université Paris Est, Créteil, FR. [89]Campbell Family Mental Health Research Institute, Centre for Addiction and Mental Health, Toronto, ON, CA. [90]Neurogenetics Section, Centre for Addiction and Mental Health, Toronto, ON, CA. [91]Department of Psychiatry, University of Toronto, Toronto, ON, CA. [92]Institute of Medical Sciences, University of Toronto, Toronto, ON, CA. [93]Department of Psychiatry, Psychosomatic Medicine and Psychotherapy, University Hospital Frankfurt, Frankfurt am Main, DE. [94]Cell Biology, SUNY Downstate Medical Center College of Medicine, Brooklyn, NY, US. [95]Institute for Genomic Health, SUNY Downstate Medical Center College of Medicine, Brooklyn, NY, US. [96]ISGlobal, Barcelona, ES. [97]Psychiatry, Altrecht, Utrecht, NL. [98]Psychiatry, GGZ inGeest, Amsterdam, NL. [99]Psychiatry, VU medisch centrum, Amsterdam, NL. [100]Psychiatry, North East London NHS Foundation Trust, Ilford, GB. [101]Clinic for Psychiatry and Psychotherapy, University Hospital Cologne, Cologne, DE. [102]Psychiatric and Neurodevelopmental Genetics Unit, Massachusetts General Hospital, Boston, MA, US. [103]HudsonAlpha Institute for Biotechnology, Huntsville, AL, US. [104]Department of Human Genetics, University of Michigan, Ann Arbor, MI, US. [105]Psychiatry, University of Illinois at Chicago College of Medicine, Chicago, IL, US. [106]Max Planck Institute of Psychiatry, Munich, DE. [107]Mental Health, NHS 24, Glasgow, GB. [108]Division of Psychiatry, Centre for Clinical Brain Sciences, University of Edinburgh, Edinburgh, GB. [109]Psychiatry, Brigham and Women's Hospital, Boston, MA, US. [110]Department of Psychiatry and Psychotherapy, University of Bonn, Bonn, DE. [111]Department of Genetics, Harvard Medical School, Boston, MA, US. [112]Department of Psychiatry, University of Michigan, Ann Arbor, MI, US. [113]Genetic Cancer Susceptibility Group, International Agency for Research on Cancer, Lyon, FR. [114]Estonian Genome Center, University of Tartu, Tartu, EE. [115]Discipline of Biochemistry, Neuroimaging and Cognitive Genomics (NICOG) Centre, National University of Ireland, Galway, Galway, IE. [116]Neuropsychiatric Genetics Research Group, Dept of Psychiatry and Trinity Translational Medicine Institute, Trinity College Dublin, Dublin, IE. [117]Institute of Neuroscience and Medicine (INM-1), Research Centre Jülich, Jülich, DE. [118]Research/Psychiatry, Veterans Affairs San Diego Healthcare System, San Diego, CA, US. [119]Department of Clinical Sciences, Psychiatry, Umeå University Medical Faculty, Umeå, SE. [120]Department of Clinical Psychiatry, Psychiatry Clinic, Clinical Center University of Sarajevo, Sarajevo, BA. [121]Department of Neurobiology, Care sciences, and Society, Karolinska Institutet and Center for Molecular Medicine, Karolinska University Hospital, Stockholm, SE. [122]Psychiatry, Harvard Medical School, Boston, MA, US. [123]Division of Clinical Research, Massachusetts General Hospital, Boston, MA, US. [124]Outpatient Clinic for Bipolar Disorder, Altrecht, Utrecht, NL. [125]Department of Psychiatry, Washington University in Saint Louis, Saint Louis, MO, US. [126]Department of Biochemistry and Molecular Biology II, Institute of Neurosciences, Center for Biomedical Research, University of Granada, Granada, ES. [127]Department of Neuroscience, Icahn School of Medicine at Mount Sinai, New York, NY, US. [128]Medicine, Psychiatry, Biomedical Informatics, Vanderbilt University Medical Center, Nashville, TN, US. [129]Department of Health Sciences Research, Mayo Clinic, Rochester, MN, US. [130]Psychiatry and Behavioral Sciences, Stanford University School of Medicine, Stanford, CA, US. [131]Rush University Medical Center, Chicago, IL, US. [132]Scripps Translational Science Institute, La Jolla, CA, US. [133]Neuroscience Research Australia, Sydney, NSW, AU. [134]Faculty of Medicine, Department of Psychiatry, School of Health Sciences, University of Iceland, Reykjavik, IS. [135]Division of Mental Health and Addiction, Oslo University Hospital, Oslo, NO. [136]NORMENT, University of Oslo, Oslo, NO. [137]Psychiatry and the Behavioral Sciences, University of Southern California, Los Angeles, CA, US. [138]Mood Disorders, PsyQ, Rotterdam, NL. [139]Institute for Medical Sciences, University of Aberdeen, Aberdeen, UK. [140]Research Division, Federal Institute for Drugs and Medical Devices (BfArM), Bonn, DE. [141]Centre for Addiction and Mental Health, Toronto, ON, CA. [142]Neurogenomics, TGen, Los Angeles, AZ,

US. [143]Psychiatry, Psychiatrisches Zentrum Nordbaden, Wiesloch, DE. [144]Computational Sciences Center of Emphasis, Pfizer Global Research and Development, Cambridge, MA, US. [145]Department of Biostatistics, Princess Margaret Cancer Centre, Toronto, ON, CA. [146]Dalla Lana School of Public Health, University of Toronto, Toronto, ON, CA. [147]Psychological Medicine, Institute of Psychiatry, Psychology & Neuroscience, King's College London, London, GB. [148]Department of Mental Health, Johns Hopkins University Bloomberg School of Public Health, Baltimore, MD, US. [149]Institute of Genetic Medicine, Johns Hopkins University School of Medicine, Baltimore, MD, US. [150]NORMENT, KG Jebsen Centre for Psychosis Research, Division of Mental Health and Addiction, Institute of Clinical Medicine and Diakonhjemmet Hospital, University of Oslo, Oslo, NO. [151]National Institute of Mental Health, Klecany, CZ. [152]Department of Psychiatry, University of Melbourne, Melbourne, Victoria, AU. [153]Department of Psychiatry and Addiction Medicine, Assistance Publique—Hôpitaux de Paris, Paris, FR. [154]Paris Bipolar and TRD Expert Centres, FondaMental Foundation, Paris, FR. [155]UMR-S1144 Team 1: Biomarkers of relapse and therapeutic response in addiction and mood disorders, INSERM, Paris, FR. [156]Psychiatry, Université Paris Diderot, Paris, FR. [157]Psychiatry, University of Pennsylvania, Philadelphia, PA, US. [158]Department of Psychiatry, University of Münster, Münster, DE. [159]Division of Endocrinology, Children's Hospital Boston, Boston, MA, US. [160]Centre for Affective Disorders, Institute of Psychiatry, Psychology and Neuroscience, London, GB. [161]Department of Psychiatry & Psychology, Mayo Clinic, Rochester, MN, US. [162]School of Medical Sciences, University of New South Wales, Sydney, NSW, AU. [163]Department of Human Genetics, University of Chicago, Chicago, IL, US. [164]Biometric Psychiatric Genetics Research Unit, Alexandru Obregia Clinical Psychiatric Hospital, Bucharest, RO. [165]Institute of Neuroscience and Physiology, University of Gothenburg, Gothenburg, SE. [166]INSERM, Paris, FR. [167]Department of Medical & Molecular Genetics, King's College London, London, GB. [168]Neuroscience Therapeutic Area, Janssen Research and Development, LLC, Titusville, NJ, US. [169]Cancer Epidemiology and Prevention, M. Sklodowska-Curie Cancer Center and Institute of Oncology, Warsaw, PL. [170]School of Psychology, The University of Queensland, Brisbane, QLD, AU. [171]Research Institute, Lindner Center of HOPE, Mason, OH, US. [172]Centre for Cognitive Ageing and Cognitive Epidemiology, University of Edinburgh, Edinburgh, GB. [173]Human Genetics Branch, Intramural Research Program, National Institute of Mental Health, Bethesda, MD, US. [174]Division of Mental Health and Addiction, Oslo University Hospital, Oslo, NO. [175]Division of Mental Health and Addiction, University of Oslo, Institute of Clinical Medicine, Oslo, NO. [176]Institute of Molecular and Cell Biology, University of Tartu, Tartu, EE. [177]Mental Health, Faculty of Medicine and Health Sciences, Norwegian University of Science and Technology—NTNU, Trondheim, NO. [178]Psychiatry, St Olavs University Hospital, Trondheim, NO. [179]Psychosis Research Unit, Aarhus University Hospital, Risskov, DK. [180]Munich Cluster for Systems Neurology (SyNergy), Munich, DE. [181]University of Liverpool, Liverpool, GB. [182]Psychiatry and Human Genetics, University of Pittsburgh, Pittsburgh, PA, US. [183]Mental Health Services in the Capital Region of Denmark, Mental Health Center Copenhagen, University of Copenhagen, Copenhagen, DK. [184]Division of Psychiatry, Haukeland Universitetssjukehus, Bergen, NO. [185]Faculty of Medicine and Dentistry, University of Bergen, Bergen, NO. [186]Human Genetics and Computational Biomedicine, Pfizer Global Research and Development, Groton, CT, US. [187]College of Medicine Institute for Genomic Health, SUNY Downstate Medical Center College of Medicine, Brooklyn, NY, US. [188]Department of Clinical Genetics, Amsterdam Neuroscience, Vrije Universiteit Medical Center, Amsterdam, NL. [189]Department of Neurology and Neurosurgery, McGill University, Faculty of Medicine, Montreal, QC, CA. [190]Montreal Neurological Institute and Hospital, Montreal, QC, CA. [191]Department of Biomedical and NeuroMotor Sciences, University of Bologna, Bologna, IT. [192]Department of Psychiatry, Massachusetts General Hospital,

Boston, MA, US. [193]Psychiatric and Neurodevelopmental Genetics Unit (PNGU), Massachusetts General Hospital, Boston, MA, US. [194]Faculty of Medicine, University of Iceland, Reykjavik, IS. [195]Department of Psychiatry, Hospital Namsos, Namsos, NO. [196]Department of Neuroscience, Norges Teknisk Naturvitenskapelige Universitet Fakultet for naturvitenskap og teknologi, Trondheim, NO. [197]Department of Genetics, University of North Carolina at Chapel Hill, Chapel Hill, NC, US. [198]Department of Psychiatry, University of North Carolina at Chapel Hill, Chapel Hill, NC, US. [199]Department of Psychiatry, McGill University, Montreal, QC, CA. [200]Dept of Psychiatry, Sankt Olavs Hospital Universitetssykehuset i Trondheim, Trondheim, NO. [201]Clinical Institute of Neuroscience, Hospital Clinic, University of Barcelona, IDIBAPS, CIBERSAM, Barcelona, ES. [202]Institute of Biological Psychiatry, MHC Sct. Hans, Mental Health Services Copenhagen, Roskilde, DK. [203]Department of Clinical Medicine, University of Copenhagen, Copenhagen, DK. [204]Psychiatry, Indiana University School of Medicine, Indianapolis, IN, US. [205]Biochemistry and Molecular Biology, Indiana University School of Medicine, Indianapolis, IN, US. [206]Department of Pathology and Laboratory Medicine, University of California Los Angeles, Los Angeles, CA, US. [†] deceased.

## Author Contributions

**Conceptualization:** Caroline C. McGrouther, Aaditya V. Rangan, Arianna Di Florio.

**Data curation:** Caroline C. McGrouther, Aaditya V. Rangan, Arianna Di Florio, John Kelsoe.

**Formal analysis:** Caroline C. McGrouther, Aaditya V. Rangan, Jeremy A. Elman.

**Funding acquisition:** Aaditya V. Rangan, Nicholas J. Schork.

**Investigation:** Caroline C. McGrouther, Aaditya V. Rangan, Arianna Di Florio.

**Methodology:** Caroline C. McGrouther, Aaditya V. Rangan, Arianna Di Florio, Jeremy A. Elman.

**Project administration:** Nicholas J. Schork, John Kelsoe.

**Resources:** Aaditya V. Rangan, Nicholas J. Schork.

**Software:** Aaditya V. Rangan.

**Supervision:** Arianna Di Florio, Nicholas J. Schork, John Kelsoe.

**Validation:** Caroline C. McGrouther, Aaditya V. Rangan, Arianna Di Florio.

**Visualization:** Caroline C. McGrouther, Aaditya V. Rangan.

**Writing – original draft:** Caroline C. McGrouther, Aaditya V. Rangan.

**Writing – review & editing:** Caroline C. McGrouther, Aaditya V. Rangan, Arianna Di Florio, Jeremy A. Elman, Nicholas J. Schork, John Kelsoe.

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
