## [Decision Letter · Decision Letter 0]

21 Aug 2024

PONE-D-24-25636Heterogeneity analysis provides evidence for a genetically homogeneous subtype of bipolar-disorderPLOS ONE

Dear Dr. Rangan,

Thank you for submitting your manuscript to PLOS ONE. After careful consideration, we feel that it has merit but does not fully meet PLOS ONE’s publication criteria as it currently stands. Therefore, we invite you to submit a revised version of the manuscript that addresses the points raised during the review process.

We look forward to receiving your revised manuscript.

Kind regards,

Austin W.T. Chiang

Academic Editor

PLOS ONE

Journal Requirements:

3. Thank you for stating the following in the Acknowledgments Section of your manuscript: "This research was supported by NIH grant 2U19AG023122-11A1".

Please remove any funding-related text from the manuscript and let us know how you would like to update your Funding Statement. Currently, your Funding Statement reads as follows: "A.R, N.S

NIH grant U19 AG023122 National Institute of Health.

www.nih.gov

The funders did not play any role in the study design, data collection and analysis, decision to publish or preparation of this manuscript."

4. Please update your submission to use the PLOS LaTeX template. The template and more information on our requirements for LaTeX submissions can be found at http://journals.plos.org/plosone/s/latex.

5. One of the noted authors is a group or consortium "Psychiatric Genomics Consortium". In addition to naming the author group, please list the individual authors and affiliations within this group in the acknowledgments section of your manuscript. Please also indicate clearly a lead author for this group along with a contact email address.

7. We notice that your supplementary figures are uploaded with the file type 'Figure'. Please amend the file type to 'Supporting Information'. Please ensure that each Supporting Information file has a legend listed in the manuscript after the references list.

8. We notice that your supplementary figures are included in the manuscript file. Please remove them and upload them with the file type 'Supporting Information'. Please ensure that each Supporting Information file has a legend listed in the manuscript after the references list.

9.Please include captions for your Supporting Information files at the end of your manuscript, and update any in-text citations to match accordingly. Please see our Supporting Information guidelines for more information: http://journals.plos.org/plosone/s/supporting-information.

Reviewers' comments:

Reviewer's Responses to Questions

**Comments to the Author**

1. Is the manuscript technically sound, and do the data support the conclusions?

Reviewer #1: Partly

Reviewer #2: Partly

Reviewer #3: Yes

2. Has the statistical analysis been performed appropriately and rigorously? 

Reviewer #1: Yes

Reviewer #2: Yes

Reviewer #3: Yes

3. Have the authors made all data underlying the findings in their manuscript fully available?

Reviewer #1: Yes

Reviewer #2: Yes

Reviewer #3: No

4. Is the manuscript presented in an intelligible fashion and written in standard English?

Reviewer #1: Yes

Reviewer #2: Yes

Reviewer #3: Yes

5. Review Comments to the Author

Reviewer #1: This paper introduces a biclustering approach to explore genetic heterogeneity in Bipolar Disorder (BD). The authors demonstrate the potential of this method to improve polygenic risk score (PRS) predictions, particularly for the BDI subtype, by identifying genetically homogeneous subsets. The key contributions of the paper include:

1. Innovative Methodology: The paper applies biclustering analysis to identify genetically homogeneous subsets within BD cases. This novel approach aims to address challenges in analyzing genetic heterogeneity that are not fully captured by existing methods.

2. Potential Applications: The study provides insights into how genetic homogeneity can be leveraged to enhance the accuracy of PRS predictions. This could lead to better understanding and prediction of BD, particularly in distinguishing between BDI and BDII subtypes.

3. Focus on Genetic Subtypes: By focusing on genetic subtypes, the paper contributes to the broader understanding of the genetic basis of BD, which could inform future research and potential therapeutic strategies.

4. Discussion on Methodological Choices: The paper raises important questions about the choice of MAF thresholds, principal component corrections, and the handling of multiple biclusters, prompting further investigation into these methodological decisions.

5. Insightful Analysis: The authors provide a detailed and thoughtful analysis of the genetic homogeneity within biclusters and explore the implications of these findings for understanding BD at a more granular level.

While the paper makes significant contributions in applying biclustering to explore genetic heterogeneity in Bipolar Disorder, I have a few questions and suggestions regarding some of the methodological choices and potential areas for further exploration.

1. MAF Threshold:

Common variants are typically defined as those with MAF ≥ 5%. Could you clarify why you selected a threshold of MAF ≥ 25%?

Is there any specific literature or reasoning supporting this higher threshold?

2. I am curious whether you have considered lowering the MAF threshold to 5% for PRS calculations. Doing so could potentially increase the SNP overlap and allow for the estimation of parameters for a larger number of SNPs. Have you attempted to present results using a lower MAF threshold, and if so, how did it impact the performance of PRSwide and PRSbicl? Would this approach potentially improve validation in Arm-3 and Arm-4?

3. Inconsistency in Principal Component Correction:

In Arm-1, you corrected for PC1 and PC2, while in Arm-2, Arm-3, and Arm-4, you corrected for PC1 through PC6 and PC19. Could you clarify why different sets of principal components were chosen for correction in these arms?

How do you ensure that these differences do not introduce biases or affect the comparability of the results across the arms?

4. Clarification on Initial Conditions:

How sensitive is the biclustering result to the initial conditions and the specific choices made during the early iterations? Did the authors test the robustness of their results by varying these conditions?

5. Overfitting Concerns: What steps did the authors take to prevent overfitting during the iterative process? How did they ensure that the identified bicluster represents true biological signal rather than noise?

6. In the PRSs section, it is mentioned that genotype-level data imputed using the 1000 Genomes European reference panel was used. However, in the DATA section, it is stated that imputed data was not retained for the primary analysis. Could you please clarify whether imputed data was used in the analysis, and if so, in which specific parts of the study? This clarification would help ensure that the methodology is fully transparent and reproducible.

7. I noticed that the algorithm mentioned in the manuscript refers to a secondary bicluster, which suggests that there might be more than one bicluster identified during the analysis. Could you clarify the following:

Did the algorithm produce multiple biclusters for the case subjects and control subjects in each iteration, or was only one primary bicluster identified for each group?

Were there any constraints or criteria that limited the size of these biclusters (in terms of the number of subjects and SNPs) as the iterations progressed?

Could you provide details on how the size of these biclusters evolved across the iterations, specifically the number of participants and the number of SNPs included in each bicluster?

8. I noticed that while arm-2 has an 85% SNP overlap with arm-1, and arm-3 and arm-4 have lower overlaps of 50% and 30% respectively, the differences in the AUC values across these arms are relatively small (with the highest AUC values being around 0.55 for arm-2 and 0.53 for arm-3 and arm-4). Could you please elaborate on why SNP overlap is emphasized as a significant contributor to the AUC differences, despite the relatively small gap in AUC values? Additionally, I noticed that in Figures S11 and S12, the -log(p-value) appears to differ significantly from what is shown in Figures 4 and 5. Do you think the different -log(p-value) would influence the conclusions you made?

9. Figures S13 and S14 do not have clearly indicated x-axis and y-axis labels. Could you please confirm if this is the case, and if so, consider adding these labels to enhance the clarity and interpretability of the figures?

10. Thank you for your insightful analysis on the genetic homogeneity observed within the identified biclusters. I am curious whether this genetic homogeneity also reflects in the subphenotypic homogeneity of the subjects. Have you explored the possibility that the genetically homogeneous subsets might also exhibit homogeneity in certain subphenotypic traits? If so, could you share any findings or insights on this aspect?

11. Thank you for your thorough and insightful analysis using the biclustering approach. I noticed that the manuscript focuses extensively on your method's application and results. I was wondering if you had considered comparing your approach with other existing methods for analyzing genetic heterogeneity? Such a comparison could potentially highlight the unique advantages or limitations of your method relative to other techniques.

Reviewer #2: The authors have derived mathematically an interesting algorithm with machine learning and without subsequent user-input to try to use genetic information of a heterogeneous form of disease (bipolar disorder) to define or derive a prediction of disease-related signals of bipolar disorder. The authors did obtain observable expression patterns in their training controls but the results from the testing controls were not as convincing.

Comments:

1. In Figure 3: After training one arm1, the replication-AUCs on arm2-4 were much lower than the training-AUC A(i) and the AUC is only 0.5. With a 0.5 AUC, this implies that the results from the testing arm was not significant. The authors claimed that the reduction of overlapped SNP was the major effect. Are there other several potential strategies to improve the replication performance, other than increasing samples size?

2. If the result in the “testing” section of the analysis wasn’t that successful in AUC 0.5, what approaches can be taken from either a clinical or data analytical method to define genetic basis that result in bipolar disorder and their subpopulations?

3. The group has classified bipolar disorder into groups I and II. However, this group of diseases also has other ‘groupings’ such as cyclothymic disorder and other specific bipolar disorder with phenomena that doesn’t fit into I or II. In the authors analyses, would they be able to apply their covariate-corrected biclustering algorithm to these groups and would these affect/change the expected results?

4. The intent of their work is to allow genetic data to define a way forward to homogenize the large variability of phenotypic data, which is a daunting task and also achieve a method to deduce disease from biclustering genetic data. Despite the interesting findings of the biclustering data, the authors should elaborate if observations they currently obtained are biologically relevant. The basis of this manuscript seems to be a “methods” developmental tool which can be used in further analysis of BD patient data and the authors should state this more clearly in the abstract and introduction.

5. The authors note that there is an enrichment of BDI versus BDII. Is there any biological basis for this that can be genetically explained? It is a little hard envision that a distinct genetic reason as compared to both epigenetic or environmental influences which will result in BDI instead of BDII.

6(i): Can the authors define the meaning of “genetic subgroups”? In biological terms, it is hard to comprehend if there are subtypes of a specific gene and/or genetic expression. Do they mean genetic variation? Or isoforms? Or perhaps “genetically stratified subgroups of individuals”?

6(ii): Last paragraph of the conclusion “sample sizes will likely be necessary to clarify the genetic- and phenotypic subtypes of BD” can be misunderstood as there might be genetic subtypes. Phenotypic subtypes can be understood from a biological standpoint but genetic subtypes (of humans) do not really make sense. Can this sentence be rephrased so it is less ambiguous?

Reviewer #3: ### Review Report: "Heterogeneity analysis provides evidence for a genetically homogeneous subtype of bipolar disorder"

(1) Relevance and Scope:

The paper falls within the scope of PLOS ONE, particularly in the domain of computational biology and mental health research. The study addresses a significant problem in the field, focusing on the genetic heterogeneity of bipolar disorder, which is relevant to both clinical applications and basic research in psychiatric genetics.

(2) Originality and Novelty:

The paper presents a novel approach to identifying a genetically homogeneous subtype of bipolar disorder using heterogeneity analysis. This approach is relatively new and has the potential to contribute to the understanding of bipolar disorder's genetic architecture. However, the authors should make it clearer which findings are new and not found in the existing literature, which findings are consistent or inconsistent with prior studies, and what issues still need to be explored in future research. This will help in highlighting the originality and significance of the study's contributions.

(3) Clarity and Structure:

The manuscript is generally clear and well-structured. The introduction effectively sets the stage for the research question, and the methodology is described in a logical sequence. However, to further improve clarity, I recommend including a flow chart of the analysis procedure. This should detail the input data, parameters to be specified, methods, evaluation indices, and output. This will help readers to better understand the steps involved in the data processing and analysis.

(4) Methodology and Algorithms:

The methodology is sound and appropriate for the research question. The authors have applied computational techniques effectively to analyze genetic data. However, there are some areas where additional detail could be provided to ensure that the computational approaches are fully transparent and reproducible. Specifically, the authors conducted bicluster analysis with covariates, and it would be beneficial to include heatmaps to illustrate these findings. These visual aids would enhance the readers' understanding of the results and their significance.

(5) Reproducibility:

While the paper provides a solid overview of the methodology, more detailed descriptions of the algorithms and data processing steps are needed to ensure reproducibility. The current statement of data availability suggests that the data can be requested from the Psychiatric Genomics Consortium. However, based on past experiences, obtaining this data can be challenging. Therefore, I recommend that the authors provide some demonstrative data that can be used for research reproducibility. This will ensure that other researchers can replicate the experiments and validate the findings.

(6) Data Analysis:

The data used in the analysis appears to be appropriate and of high quality. The authors have chosen relevant datasets to explore the genetic heterogeneity of bipolar disorder. However, the paper could benefit from a more detailed discussion of the data selection process and any potential biases or limitations inherent in the data.

(7) Statistical Analysis:

The statistical analysis is generally sound, with appropriate methods used to validate the findings. However, the paper would benefit from a more thorough explanation of the statistical techniques employed and a clearer justification for their use. Additionally, the authors should consider performing additional validation steps to strengthen the robustness of their conclusions.

(8) Results and Interpretation:

The results are presented clearly and are well-supported by the data. The interpretation of the results is logical, and the authors make a convincing case for the existence of a genetically homogeneous subtype of bipolar disorder. However, as mentioned earlier, providing heatmaps of the bicluster analysis results would greatly enhance the interpretation of the findings. Additionally, the authors should be explicit about which findings are new and how they compare to existing literature.

(9) Discussion and Conclusion:

The discussion provides a good overview of the study's findings and their relevance to the field. However, it could be expanded to include a more in-depth exploration of the study's limitations and how they might be addressed in future research. The conclusion is concise and effectively summarizes the key points but should also consider the broader impact of the findings on the field.

(10) Ethical Considerations:

The paper does not raise any immediate ethical concerns related to data collection or the use of human subjects. However, it would be helpful if the authors explicitly stated any ethical approvals obtained for the use of genetic data and addressed any potential privacy concerns.

(11) Significance and Impact:

The research has the potential to make a significant impact on the field of psychiatric genetics by identifying a genetically homogeneous subtype of bipolar disorder. This finding could lead to more targeted treatments and a better understanding of the genetic basis of the disorder. The study's impact could be enhanced by providing more detail on how these findings might be applied in clinical practice.

(12) Language and Writing:

The paper is well-written, with clear and concise language. The grammar and expression are generally good, but a few minor edits could improve readability. The authors should ensure that all technical terms are adequately defined for readers who may not be experts in the field.

Overall, this paper presents an interesting and potentially impactful study on the genetic heterogeneity of bipolar disorder. With these revisions to enhance clarity, reproducibility, and depth of discussion, it could make a valuable contribution to the field.

6. PLOS authors have the option to publish the peer review history of their article (what does this mean?). If published, this will include your full peer review and any attached files.

Reviewer #1: No

Reviewer #2: No

Reviewer #3: No

---

## [Author Response · Author response to Decision Letter 0]

8 Oct 2024

We have uploaded the response to the editor and the reviewers as a pdf.

---

## [Decision Letter · Decision Letter 1]

8 Nov 2024

Heterogeneity analysis provides evidence for a genetically homogeneous subtype of bipolar-disorder

PONE-D-24-25636R1

Dear Dr. Rangan,

We’re pleased to inform you that your manuscript has been judged scientifically suitable for publication and will be formally accepted for publication once it meets all outstanding technical requirements.

Kind regards,

Wan-Tien Chiang

Academic Editor

PLOS ONE

Additional Editor Comments (optional):

Reviewers' comments:

Reviewer's Responses to Questions

**Comments to the Author**

1. If the authors have adequately addressed your comments raised in a previous round of review and you feel that this manuscript is now acceptable for publication, you may indicate that here to bypass the “Comments to the Author” section, enter your conflict of interest statement in the “Confidential to Editor” section, and submit your "Accept" recommendation.

Reviewer #1: All comments have been addressed

Reviewer #2: All comments have been addressed

2. Is the manuscript technically sound, and do the data support the conclusions?

Reviewer #1: Yes

Reviewer #2: (No Response)

3. Has the statistical analysis been performed appropriately and rigorously? 

Reviewer #1: Yes

Reviewer #2: (No Response)

4. Have the authors made all data underlying the findings in their manuscript fully available?

Reviewer #1: Yes

Reviewer #2: (No Response)

5. Is the manuscript presented in an intelligible fashion and written in standard English?

Reviewer #1: Yes

Reviewer #2: (No Response)

6. Review Comments to the Author

Reviewer #1: 1.2 Thank you for the detailed response. Could you please add a brief explanation in the main text on why the MAF threshold of >25% was chosen? This would provide readers with clarity on this methodological choice. Thank you!

3. Thank you for clarifying the principal component correction approach and for providing additional details in the main text and S1 Text. This clarification helps ensure the consistency and comparability of results across different arms.

4.5.

Thank you for explaining the design of your sensitivity analysis approach and the steps taken to prevent overfitting. I appreciate the thorough testing of the biclustering method’s robustness and the safeguards you've incorporated, as well as the additional details provided in S1 Text and the main text, including the new Fig 11. These clarifications effectively strengthen and clarify your methodology.

6.

Thank you for the clarification regarding the use of imputed and raw data in different parts of the analysis. I appreciate the updates in the main text and S1 Text, which help enhance the transparency and reproducibility of your methodology.

7.

Thank you for the comprehensive clarification on the generation of multiple biclusters and the criteria for significant biclusters. I appreciate the details on the constraints applied to the bicluster size as iterations progress and the rationale behind focusing on the case-bicluster in Arm-1, given the heterogeneity concerns in the controls for other arms. Including this information in S1 Text significantly enhances the clarity of your approach.

8.

I appreciate the detailed explanation regarding the impact of SNP overlap on AUC differences and the clarification on why structural differences in specific genotyping platforms might lead to real data results differing from simulated outcomes. Additionally, could you please include a brief discussion in the Discussion section on the observed differences in -log(p-value) between Figures S11, S12 and Figures 4, 5? This would help clarify whether these differences might impact the consistency or stability of the conclusions. Thank you!

9.

Thank you for addressing this. I appreciate the addition of the axis labels to enhance clarity in Figures 15 and 16 in S1 Text.

10. 11.

Thank you for the detailed responses. I appreciate the consideration of genetic homogeneity and subphenotypic traits and understand the current limitations due to data access. Additionally, your comparison of the biclustering method to UMap and Louvain-clustering provides valuable insight into its strengths and limitations. Including these discussions in S1 Text enhances the transparency of your approach and highlights its unique contributions. Thank you for addressing these points.

Reviewer #2: (No Response)

7. PLOS authors have the option to publish the peer review history of their article (what does this mean?). If published, this will include your full peer review and any attached files.

Reviewer #1: No

Reviewer #2: No

---

## [Editor Report · Acceptance letter]

13 Nov 2024

PONE-D-24-25636R1 

PLOS ONE

Dear Dr. Rangan, 

I'm pleased to inform you that your manuscript has been deemed suitable for publication in PLOS ONE. Congratulations! Your manuscript is now being handed over to our production team.

Kind regards, 

on behalf of

Dr. Wan-Tien Chiang 

Academic Editor

PLOS ONE